# Mechanism of filament formation in UPA-promoted CARD8 and NLRP1 inflammasomes

L. Robert Hollingsworth [1,2,3,11], Liron David[1,2,11], Yang Li[4,11], Andrew R. Griswold[5,6], Jianbin Ruan[1,2,7], Humayun Sharif[1,2], Pietro Fontana[1,2], Elizabeth L. Orth-He[8], Tian-Min Fu [1,2,9], Daniel A. Bachovchin [6,8,10] & Hao Wu [1,2,3✉]

NLRP1 and CARD8 are related cytosolic sensors that upon activation form supramolecular signalling complexes known as canonical inflammasomes, resulting in caspase−1 activation, cytokine maturation and/or pyroptotic cell death. NLRP1 and CARD8 use their C-terminal (CT) fragments containing a caspase recruitment domain (CARD) and the UPA (conserved in UNC5, PIDD, and ankyrins) subdomain for self-oligomerization, which in turn form the platform to recruit the inflammasome adaptor ASC (apoptosis-associated speck-like protein containing a CARD) or caspase-1, respectively. Here, we report cryo-EM structures of NLRP1-CT and CARD8-CT assemblies, in which the respective CARDs form central helical filaments that are promoted by oligomerized, but flexibly linked, UPAs surrounding the filaments. Through biochemical and cellular approaches, we demonstrate that the UPA itself reduces the threshold needed for NLRP1-CT and CARD8-CT filament formation and signalling. Structural analyses provide insights on the mode of ASC recruitment by NLRP1-CT and the contrasting direct recruitment of caspase-1 by CARD8-CT. We also discover that subunits in the central NLRP1$^{CARD}$ filament dimerize with additional exterior CARDs, which roughly doubles its thickness and is unique among all known CARD filaments. Finally, we engineer and determine the structure of an ASC$^{CARD}$–caspase-1$^{CARD}$ octamer, which suggests that ASC uses opposing surfaces for NLRP1, versus caspase-1, recruitment. Together these structures capture the architecture and specificity of the active NLRP1 and CARD8 inflammasomes in addition to key heteromeric CARD-CARD interactions governing inflammasome signalling.

[1] Department of Biological Chemistry and Molecular Pharmacology, Harvard Medical School, Boston, MA 02115, USA. [2] Program in Cellular and Molecular Medicine, Boston Children's Hospital, Boston, MA 02115, USA. [3] Program in Biological and Biomedical Sciences, Harvard Medical School, Boston, MA 02115, USA. [4] Department of Biophysics, University of Texas Southwestern Medical Center, Dallas, TX 75390, USA. [5] Weill Cornell/Rockefeller/Sloan Kettering Tri-Institutional MD-PhD Program, New York, NY, USA. [6] Pharmacology Program, Weill Cornell Graduate School of Medical Sciences, Memorial Sloan Kettering Cancer Center, New York, NY 10065, USA. [7] Department of Immunology, University of Connecticut Health Center, Farmington, CT, USA. [8] Tri-Institutional PhD Program in Chemical Biology, Memorial Sloan Kettering Cancer Center, New York, NY 10065, USA. [9] Department of Biological Chemistry and Pharmacology, Comprehensive Cancer Center, The Ohio State University, Columbus, OH, USA. [10] Chemical Biology Program, Memorial Sloan Kettering Cancer Center, New York, NY, USA. [11]These authors contributed equally: L. Robert Hollingsworth, Liron David, Yang Li. ✉email: wu@crystal.harvard.edu

nnate immune pathways recognize and respond to a diverse array of intracellular threats. In one such pathway, cells identify and amplify danger signals through supramolecular signaling complexes called canonical inflammasomes[1,2]. Upon recognition of intracellular molecular patterns indicative of pathogens or endogenous damage, sensor proteins facilitate inflammasome assembly by undergoing large conformational changes that lead to their oligomerization[3,4]. Nucleotide binding domains (NBDs) often drive this self-oligomerization to cluster death-fold domains[5,6], which in turn recruit downstream adapter and effector molecules. These death-fold domains, including pyrin domains (PYD) and caspase recruitment domains (CARD), participate in homotypic (CARD–CARD or PYD–PYD) interactions that ultimately lead to polymerization of caspase-1 into filaments[3,4,7–9]. This process increases the local concentration of caspase catalytic domains to facilitate its homo-dimerization and autoproteolysis, resulting in its activation[1–3]. Active caspase-1, a cysteine protease, then processes pro-inflammatory cytokines, including pro-IL-1β and pro-IL-18, to their bioactive forms and cleaves the pore-forming protein gasdermin-D (GSDMD) to promote cytokine release and often concomitant lytic cell death termed pyroptosis[10–12].

NLRP1 and CARD8, two related inflammasome sensors, are highly expressed in a number of cell types and play important roles in both host defense and human diseases. Keratinocytes constitutively express high levels of NLRP1 and germline mutations of NLRP1 in humans lead to a number of skin-related inflammatory diseases, including multiple self-healing palmoplantar carcinoma (MSPC), familial keratosis lichenoides chronica (FKLC)[13], vitiligo[14,15], autoinflammation with arthritis and dyskeratosis (AIADK)[16,17], and juvenile-onset recurrent respiratory papillomatosis (JRRP)[18]. These mutations cause constitutive NLRP1 activation and downstream pyroptosis[13,16], leading to damaging inflammation. For CARD8, the most highly expressing cell types are hematopoietic in origin[19–21], and CARD8 activators are being pursued for treatment of acute myeloid leukemia[19]. Thus, understanding the molecular mechanisms governing NLRP1 and CARD8 inflammasome signaling will facilitate the discovery of new therapeutics for inflammatory diseases and cancers.

NLRP1 and CARD8 mediate inflammasome formation through their CARD-containing C-terminal fragment (CT) generated upon functional degradation of their respective N-terminal fragment (NT) by the proteasome[22,23]. This unusual mechanism of inflammasome activation is associated with autoproteolysis of the function-to-find domain (FIIND) common to NLRP1 and CARD8, which splits FIIND into ZU5 (found in ZO-1 and UNC5) and UPA (conserved in UNC5, PIDD, and ankyrins) subdomains and results in noncovalently associated NT and CT[24,25] (Fig. 1a). NLRP1-CT and CARD8-CT are therefore repressed by the NT until upstream cues induce proteasomal degradation of the NT and concomitant release of the CT[13,26–28] (Fig. 1b). Interestingly, one such cue is provided by small-molecule inhibitors of dipeptidyl peptidases DPP9 and DPP8, for which the mechanism of NLRP1 and CARD8 activation is currently unclear[16,19,29]. Activated CARD8 and NLRP1 inflammasomes are distinctive in that they are only composed of UPA and CARD, unlike others such as NLRC4 and NLRP3 inflammasomes which use NBDs and leucine-rich repeats (LRRs) to facilitate oligomerization and inflammasome activation[4–6,30].

In addition to having the unique UPA subdomain, the specificity of NLRP1-CT and CARD8-CT for the CARD and PYD-containing adapter ASC (apoptosis-associated speck-like protein containing a CARD) and for caspase-1 differ from most other inflammasome sensors (Fig. 1c). While ASC is a nearly universal adapter that bridges interactions between sensors and the CARD-containing caspase-1[1,31], CARD8 does not interact with ASC and instead directly engages caspase-1[32]. In contrast to the CARD-containing sensor protein NLRC4, which can activate caspase-1 both with and without ASC[6,33,34], human NLRP1 is completely dependent on ASC and cannot engage caspase-1 directly[32].

The molecular bases for the assembly of UPA-CARD-mediated NLRP1-CT and CARD8-CT inflammasomes and for their differential specificity to ASC and caspase-1 are unknown. Here, we determined the cryo-electron microscopy (cryo-EM) structures of NLRP1-CT and CARD8-CT filaments and analysed their assembly and specificity. Despite being invisible in the cryo-EM densities, the UPA domain is required for productive NLRP1 and CARD8 inflammasome signaling, suggesting that it promotes CARD clustering and filament formation and serves an analogous function to the NBD and LRRs in many other inflammasome sensors. The CARD filament of CARD8-CT resembles the helical filaments of caspase-1[CARD], ASC[CARD], and NLRC4[CARD][7,35,36] with certain structural variations. Surprisingly however, the CARD filament of NLRP1-CT is composed of CARD dimers in which an additional CARD flanks each subunit of the central CARD filament. We also determined the structure of an ASC-[CARD]-caspase-1[CARD] octamer, which suggests that ASC uses opposing surfaces for caspase-1 versus NLRP1 engagement and suggests a hierarchical inflammasome assembly mechanism. In sum, we discover new mechanisms of inflammasome formation and uncover the structural basis of hetero-oligomeric CARD–CARD interactions.

## Results

**Cryo-EM structure determination of CARD8 and NLRP1 CARD filaments.** In general, inflammasomes leverage supramolecular filamentous structures to nucleate the polymerization of caspase-1, which in turn increases the local concentration of its caspase domain to facilitate dimerization and activation[3,7,36]. To elucidate if and how CARD8-CT and NLRP1-CT form filaments, we expressed these CTs in fusion with an N-terminal maltose-binding protein (MBP) tag separated by a linker cleavable by the human rhinovirus (HRV) 3C protease. We posited that such a bulky tag would disrupt oligomerization and facilitate purification of monomeric proteins for controlled CT filament formation in vitro (Fig. 1b and Supplementary Figs. 1, 2). However, the MBP-fusion protein still formed small oligomers. By systematically optimizing cleavage conditions to minimize aggregation, we purified short (~100–200 nm) filaments, which behaved well on cryo-EM grids and enabled the calculation of CARD8-CT and NLRP1-CT maps at resolutions of 3.3 Å and 3.6 Å, respectively (Fig. 1d–i and Supplementary Figs. 1, 2, Supplementary Table 1). Unexpectedly, 2D classifications and 3D reconstructions revealed both similarities and differences between these two filament structures (Fig. 1d, i).

**Structure of the CARD8-CT filament.** Despite being included in the construct (Supplementary Fig. 1b, c), no UPA density was observed in the CARD8-CT cryo-EM reconstruction, which instead only reveals the central helical CARD filament (Fig. 2a, b). One possible explanation is that UPA does not follow the CARD helical symmetry but is orderly associated with the central CARD filament. However, subtracting out the central density followed by 3D classification without imposing symmetry did not reveal any new density. Given the predicted 17-residue unstructured linker between UPA and CARD, we proposed that UPA and any residual MBP molecules must be flexibly linked to the core CARD filament. Thus, their stochastic position relative to the central CARD filament caused them to average out during data processing. In fact, either the UPA itself or any residual uncleaved MBP

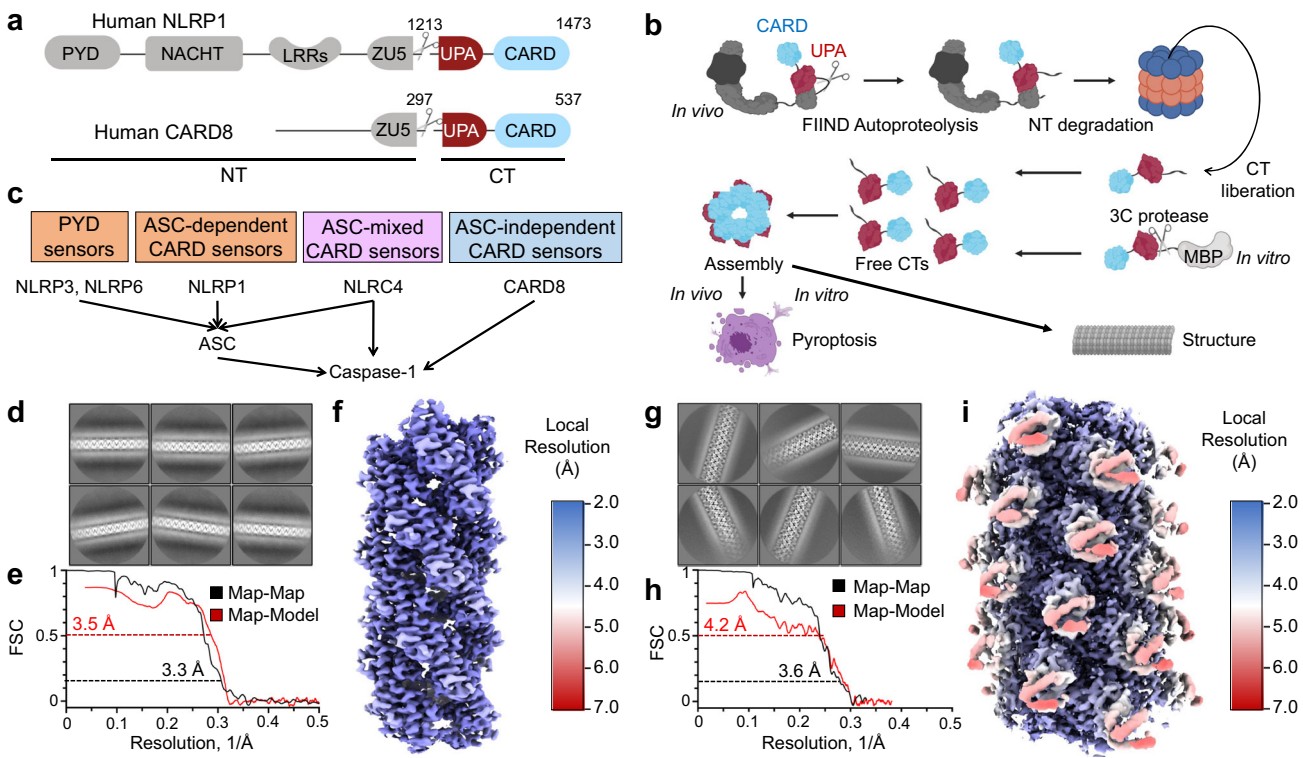

**Fig. 1 Structure determination of CARD8-CT and NLRP1-CT filaments. a** Domain architecture of human NLRP1 and CARD8, each with a C-terminal domain (CT) containing UPA-CARD. **b** CARD8 and NLRP1 activation pathway in which signals leading to NLRP1 and CARD8 activation target them to the proteasome for N-terminal degradation. This leads to the release of free CT oligomers, which assemble the inflammasome. The purification strategy bypasses N-terminal degradation by directly expressing MBP-CT fusion proteins and cleaving the tag in vitro. **c** Classification of different inflammasome sensors by their signaling through the adapter ASC. Human NLRP1 and CARD8 are ASC-dependent and ASC-independent, respectively. **d** Cryo-EM 2D classification of CARD8-CT filaments. **e** Gold-standard Fourier shell correlation (FSC) and map-model correlation plots of the CARD8-CT filament 3D reconstruction, which gave an overall resolution of 3.3 Å. **f** Local resolution of the CARD8-CT filament calculated with RELION's local resolution estimation[59,70] and colored as indicated. **g** Cryo-EM 2D classification of NLRP1-CT filaments. **h** FSC plots of the NLRP1-CT filament 3D reconstruction, which gave an overall resolution of 3.6 Å. **i** Local resolution of the NLRP1-CT filament with RELION's local resolution estimation[59,70] and colored as indicated.

tag appeared as noisy density surrounding the filaments in 2D class averages (Fig. 1d).

Like other CARD filament structures[4,7,35–38], the CARD8$^{CARD}$ filament also possesses a one-start helical symmetry, with a refined left-handed rotation of 99.1° and an axial rise of 5.2 Å per subunit (~3.6 subunits per helical turn). The filament has a diameter of ~80 Å with a central hole of ~10 Å (Fig. 2a, b). Analysis of the three types of asymmetric interactions (Fig. 2c) that are characteristic of death-fold filaments[39] revealed that the type I interaction is unusually small, with only ~150 Å$^2$ buried surface area per partner, in comparison to ~350 and ~560 Å$^2$ for those in ASC$^{CARD}$ and caspase-1$^{CARD}$ filaments[7,36]. In contrast, the type II interface is substantially more extensive, burying ~490 Å$^2$ surface area per partner, and the type III interface covers ~230 Å$^2$ per partner. Because of the small type I interface, the filament structure looks almost perforated with visible gaps (Fig. 2a).

Detailed inspection of the three interfaces revealed many charge–charge pairs as well as other hydrophilic interactions including R459 at type Ia with D473 and D477 at type Ib, K509 of type IIa with D525 of type IIb, E507 of type IIa with R464 of type IIb, E479 of type IIa with Y527 of type IIb, and E483 and E487 of type IIIa with R495 of IIIb (Fig. 2d). We used these structural insights to design point mutations that would abolish CARD filament formation. To assess the impact of these mutations on filament formation, we cleaved recombinant wild-type and

mutant MBP-tagged CARD8$^{CARD}$ proteins for 3 h at 37 °C, followed by EM imaging of the negatively stained samples. While the wild-type protein formed filaments when the bulky MBP tag was removed, seven different point mutants, one at the type I interface, four at the type II interface, and two at the type III interface, completely abolished filament formation in vitro (Fig. 2e). Of note, unlike many CARDs that form filaments at low µM concentrations[7,36,38], we had to raise the concentration of CARD8$^{CARD}$ to 15 µM to see consistent filaments in multiple grid areas under negative stain EM.

Next, we employed a reconstituted HEK293T cell system stably expressing caspase-1 and GSDMD to assess whether these mutants were able to signal in cells[19]. We transfected these cells with plasmid encoding for WT or mutant CARD8$^{UPA-CARD}$, and 24 h later, analysed the supernatant for lactate dehydrogenase (LDH) activity, a hallmark of pyroptotic cell death, and the lysate by immunoblotting for caspase-1 and GSDMD cleavage (Fig. 2f). As expected, WT CARD8$^{UPA-CARD}$ induced elevated LDH activity in the supernatant and exhibited prominent caspase-1 and GSDMD cleavage. In contrast, five out of the six filament-deficient mutants showed background level LDH release with no discernible caspase-1 or GSDMD cleavage, similar to the empty vector (EV). Only the D511K mutant displayed residual caspase-1 and GSDMD cleavage. Of note, expression of the CARD8$^{CARD}$ itself did not lead to inflammasome signaling under these same conditions (Fig. 2f).

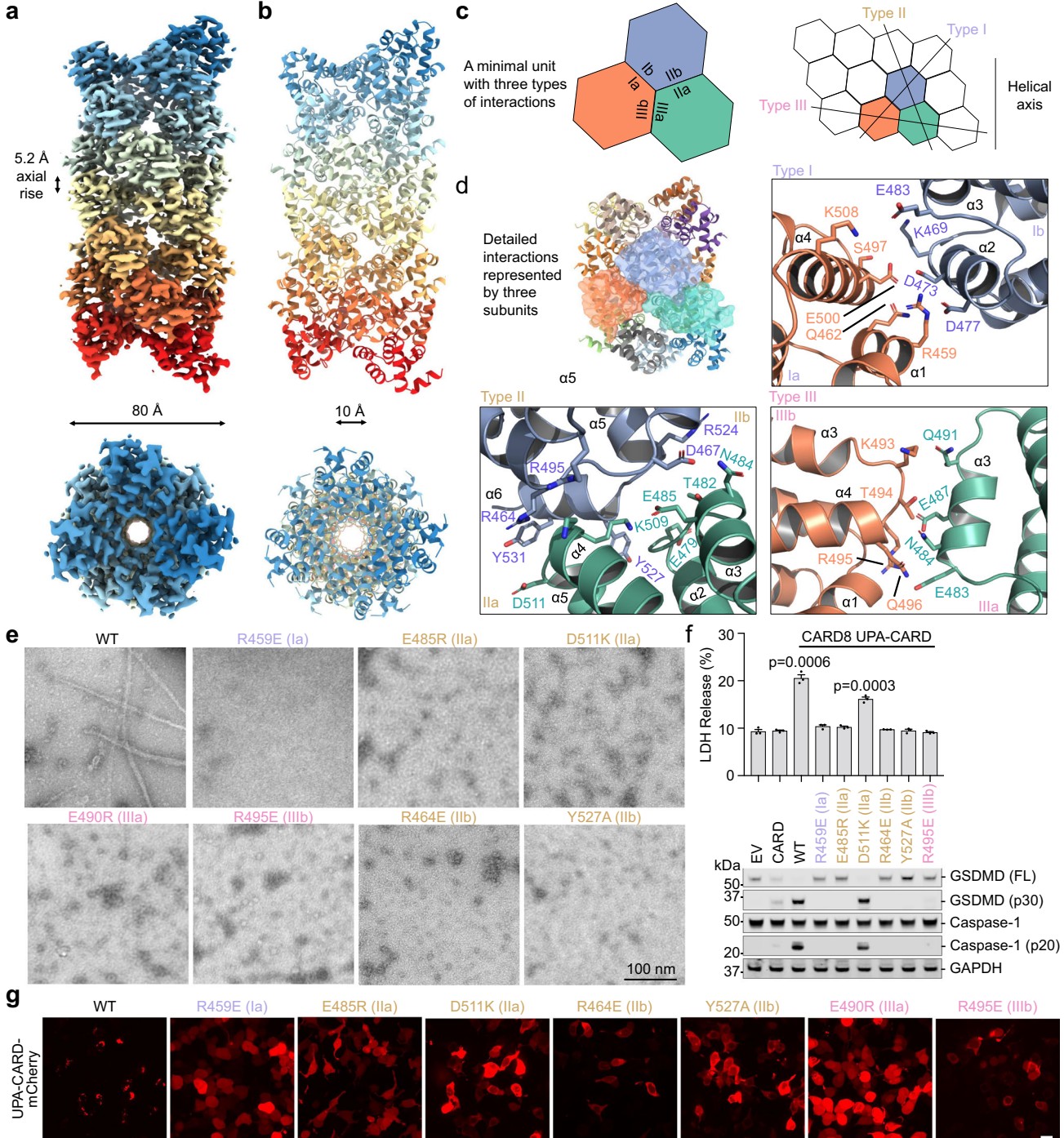

**Fig. 2 Cryo-EM structure and mutagenesis of the CARD8-CT filament. a, b** Overall structure of the CARD8-CT inflammasome, with a resolved CARD8[CARD] filament. The cryo-EM density (**a**) and atomic model (**b**) are colored by subunit. **c** Illustration of classical CARD filament interfaces: type I, II, and III. **d** Detailed CARD–CARD interactions in the CARD8-CT filament structure colored as in **c**. **e** Negative stain electron microscopy snapshots of CARD8[CARD] wild-type (WT) and mutants that abolish filament formation in vitro. Filaments were formed at 15 μM (shown here) and 30 μM. Micrographs are representative of >3 fields of view for each sample. **f** Inflammasome assay of CARD8-CT-mCherry mutants transiently expressed in HEK293T cells that stably express caspase-1 and GSDMD. Filament-deficient mutants abolish inflammasome activity as measured by LDH release (top) and western blot (bottom) for activated inflammasome components. *P*-values are compared to empty vector (EV) by two-sided Student's *t*-test. Exact *P*-values are provided in the Source Data File. Data are means ± SEM of n = 3 biological replicates. **g** Confocal imaging of HEK293T cells transiently expressing CARD8[UPA-CARD]-mCherry WT and mutants. Filament-deficient mutants abolish filament formation in cells. Experiments were performed with two biological replicates. Scale bar 5 μm. In **e–g**, mutations are labeled in the colors of the interface types as defined in **c**.

We further expressed CARD8[UPA-CARD] fused to a C-terminal mCherry tag in HEK293T cells and found that although the WT formed strong punctate structures indicative of filament formation, all the CARD8[UPA-CARD] constructs with CARD interface mutations showed diffuse distributions consistent with defective filament formation (Fig. 2g). Thus, these cellular data validated the structure and confirmed that filamentous CARD–CARD interactions are crucial for CARD8-CT-mediated inflammasome signaling.

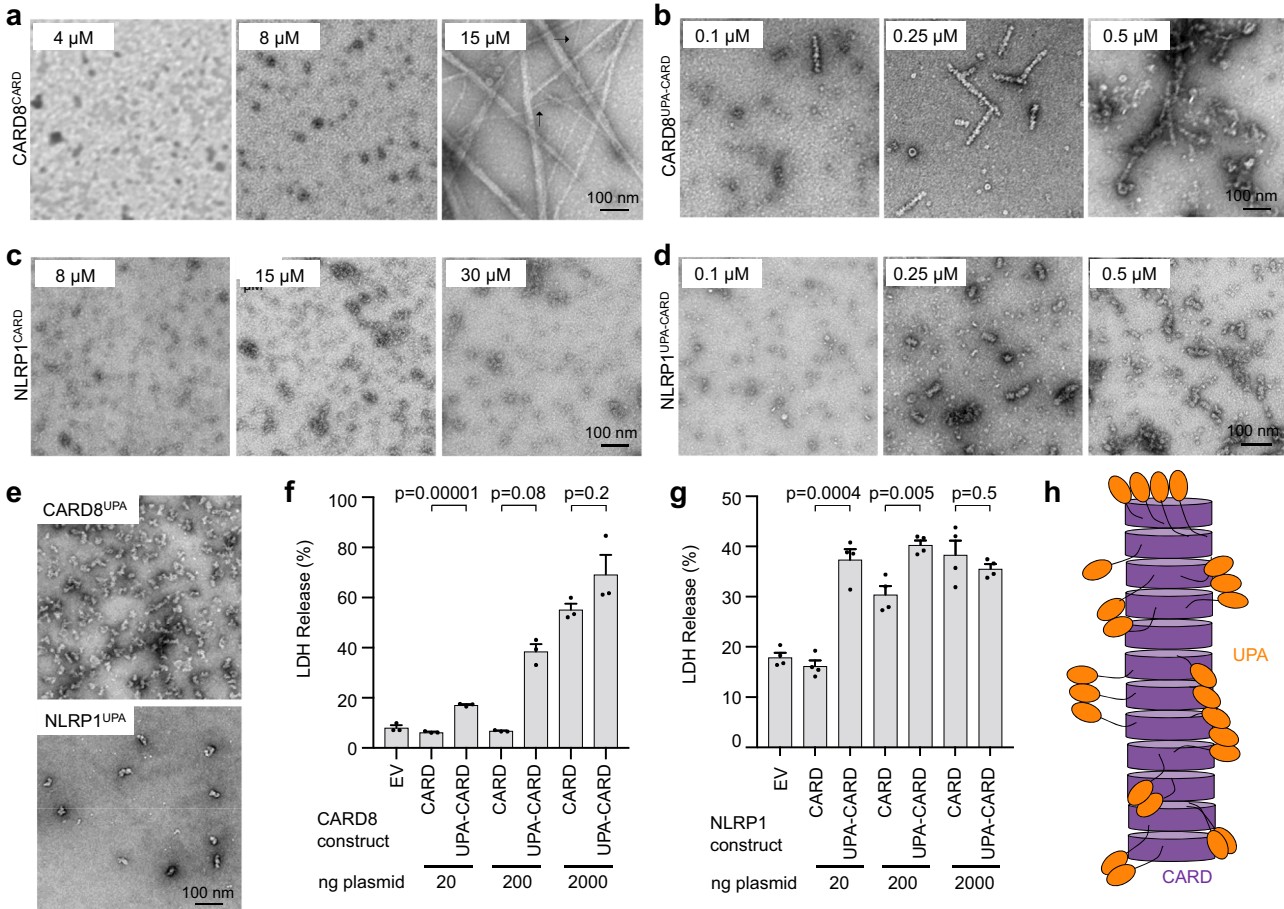

**Fig. 3 The UPA subdomain reduces the concentration required for inflammasome formation. a**, **b** Concentration-dependent formation of CARD8$^{CARD}$ (**a**) and CARD8$^{UPA-CARD}$ (**b**) filaments, visualized by negative staining EM. Purified monomeric MBP-fused proteins were cleaved at concentrations ranging from 0.1 to 15 μM. CARD8$^{CARD}$ filaments did not appear consistently until 15 μM. CARD8$^{UPA-CARD}$ filaments formed consistently at 0.25 μM. Arrows in **a** indicates single, unbundled filaments. **c**, **d** Concentration-dependent formation of NLRP1$^{CARD}$ (**c**) and NLRP1$^{UPA-CARD}$ (**d**) filaments. Purified monomeric MBP-fused proteins were cleaved at concentrations ranging from 0.1 to 30 μM. NLRP1$^{CARD}$ filaments did not appear even at 30 μM, whereas NLRP1$^{UPA-CARD}$ filaments formed consistently at 0.25 μM. For **a**–**d**, filament formation was performed with two biological replicates. Micrographs are representative of >3 fields of view for each condition. **e** CARD8$^{UPA}$ and NLRP1$^{UPA}$ imaged by negative staining EM, representative of >5 fields of view. **f** Titration of CARD8$^{CARD}$ and CARD8$^{UPA-CARD}$ in HEK293T cells stably expressing caspase-1 and GSDMD. A ~100-fold higher amount of the CARD8$^{CARD}$ plasmid was needed to achieve a comparable level of cell death (marked by LDH release) by CARD8$^{UPA-CARD}$. Data are means ± SEM of $n = 3$ biological replicates. **g** Titration of NLRP1$^{CARD}$ and NLRP1$^{UPA-CARD}$ with ASC co-expression in HEK293T cells stably expressing caspase-1 and GSDMD. A ~100-fold higher amount of the NLRP1$^{CARD}$ plasmid was needed to achieve a comparable level of cell death (marked by LDH release) by NLRP1$^{UPA-CARD}$. *P*-values by two-sided Student's *t*-test. Exact *P*-values are provided in the Source Data File. Data are means ± SEM of $n = 3$ biological replicates. EV empty vector. **h** Model of UPA-enhanced inflammasome formation. The UPA subdomain increases the multivalency of the CARD8 and NLRP1 inflammasomes, lowering the concentration required to nucleate filaments and subsequently signal.

**UPA enhances NLRP1-CT and CARD8-CT signaling.** Since the CARD8$^{CARD}$ construct itself did not cause cell death while CARD8$^{UPA-CARD}$ did (Fig. 2f), we hypothesized that the presence of the UPA subdomain enhances inflammasome formation. To test this hypothesis, we sought to determine the approximate minimum concentration required to form CARD8$^{CARD}$ versus CARD8$^{UPA-CARD}$ filaments in vitro. CARD8$^{CARD}$ did not form filaments until 15 μM, while only 0.25 μM was required for CARD8$^{UPA-CARD}$ filament assembly (Fig. 3a, b), an almost 100-fold difference. We found that NLRP1$^{CARD}$ did not form any filaments even at 30 μM, but like CARD8$^{UPA-CARD}$, NLRP1$^{UPA-CARD}$ formed filaments at 0.25 μM (Fig. 3c, d), suggesting that the UPA subdomain significantly decreased the concentration needed to form CARD8 or NLRP1 filaments. Consistent with their ability to promote filament formation, CARD8$^{UPA}$ and NLRP1$^{UPA}$ alone showed aggregation under negative stain EM (Fig. 3e).

Since CARD8$^{CARD}$ formed filaments at very high concentrations in vitro, we posited that higher concentrations of transfected plasmids were required for appreciable cell signaling. To determine whether increasing the amount of transfected plasmid could lead to inflammasome formation and LDH release for the CARD8$^{CARD}$ or NLRP1$^{CARD}$ alone constructs, we raised the plasmid amount from the usual 20 to 200 ng and 2000 ng per transfection (Fig. 3f, g). For CARD8$^{CARD}$, significant induction of LDH release, and thus cell death, only occurred when 2000 ng of plasmid was used, and at this condition, CARD8$^{UPA-CARD}$ caused a similar level of LDH release (Fig. 3f). For NLRP1$^{CARD}$, 200 ng per transfection led to significant LDH release, and at 2000 ng per transfection, it triggered similar levels of LDH release to NLRP1$^{UPA-CARD}$ (Fig. 3g). These data suggest that a ~100-fold greater plasmid amount is needed to cause a similar level of CARD-mediated cell death compared to UPA-CARD-mediated

cell death. Additionally, we and others recently communicated pre-prints on the cryo-EM structures of the NLRP1-DPP9 complex, which revealed a front-to-back UPA oligomerization interface required for inflammasome signaling[40,41]. Taken together with these results, we conclude that UPA oligomerization promotes CARD8-CT and NLRP1-CT filament formation and subsequent inflammasome signaling (Fig. 3h).

**Structure of the NLRP1-CT filament containing CARD dimers**. Similar to the CARD8-CT filament structure, no UPA density was observed in the NLRP1-CT cryo-EM map, suggesting that the UPA subdomain, with the 25-residue predicted unstructured linker to the CARD, is also flexibly connected to the core NLRP1$^{CARD}$ filament (Fig. 4a and Supplementary Fig. 2b). The NLRP1$^{CARD}$ filament has one-start helical symmetry, with a left-handed rotation of 100.8° and an axial rise of 5.1 Å per subunit (~3.6 subunits per helical turn), similar to other CARD filament structures[4,7,35–38]. Strikingly however, unlike any other CARD filament structures known to date, the NLRP1$^{CARD}$ filament is composed of NLRP1$^{CARD}$ dimers, rather than monomers (Fig. 4a, b), which is also reflected in the thicker dimensions of NLRP1-CT filaments compared to CARD8-CT filaments in both 2D classes and the 3D volume (Fig. 1d, f, g, i). The inner NLRP1$^{CARD}$ filament is roughly equivalent to other CARD filament structures, and for all NLRP1$^{CARD}$ subunits in the inner filament, the dimerically related NLRP1$^{CARD}$ subunits form the outer layer of the NLRP1$^{CARD}$ filament (Fig. 4a). The total diameter of the NLRP1$^{CARD}$ filament is approximately 140 Å, with an inner hole of ~10 Å (Fig. 4a).

The NLRP1$^{CARD}$ dimer is mediated by reciprocal interactions at helices α5 and α6, burying a substantial ~300 Å$^2$ surface area per partner, and involving residues with large side chains such as Y1445, M1457, W1460, and E1461 (Fig. 4b). These regions of helices α5 and α6 are not involved in the inner core filament interaction, which instead is mediated by the classical type I, II, and III CARD–CARD interactions (Fig. 4c). Different from the CARD8$^{CARD}$ filament, the type I interface in the NLRP1$^{CARD}$ filament is more extensive, with type I, II and III interfaces burying ~300, 430, and 140 Å$^2$ surface area per partner, respectively. Charged and hydrophilic interactions dominate the interactions, including R1386 of type Ia with D1401, Y1413, and H1404 of type Ib, R1427 of type Ia with E1397 and D1401 of type Ib, E1411 of type IIa with S1395 of type IIb, Q1434 and D1437 of type IIa with R1392 of type IIb, and E1414 of type IIIa with T1421 of type IIIb (Fig. 4c).

To elucidate the role of the observed NLRP1$^{CARD}$ filament in signaling, we employed the same reconstituted HEK293T cell system stably expressing caspase-1 and GSDMD used for assessing CARD8 mutants in cells (Fig. 2f). We transfected these cells with plasmids encoding ASC and either WT or mutant NLRP1$^{UPA-CARD}$ and 24 h later, analysed the supernatant for LDH activity, and the lysate for caspase-1 and GSDMD cleavage (Fig. 4d). In contrast to WT NLRP1$^{UPA-CARD}$, structure-designed mutants including R1427E (Ia), E1397R (Ib), D1401R (Ib), E1411R (IIa), R1392E (IIb), and E1414R (IIIa) compromised cell death measured by both LDH release and caspase-1 processing (Fig. 4d). Together, the mutational analysis demonstrated the validity of the filament structure in NLRP1 signaling.

**NLRP1$^{CARD}$ has a propensity for dimerization**. To investigate the function of NLRP1$^{CARD}$ dimerization, we compared the signaling activity of WT NLRP1$^{UPA-CARD}$ and NLRP1$^{UPA-CARD}$ with several CARD dimerization mutants (Fig. 4d). Among the NLRP1$^{UPA-CARD}$ constructs with mutations on the CARD dimer interface, Y1445A compromised LDH release but retained

caspase-1 processing, suggesting partial defectiveness. However, several other dimerization mutants, including M1457A, W1460A, and E1461R, showed no discernible impact on inflammasome signaling (Fig. 4d and Supplementary Fig. 3a). Furthermore, while the Y1445A mutant abolished NLRP1$^{UPA-CARD}$ filament formation in vitro, the other single mutants still formed filaments (Supplementary Fig. 3b). These results are consistent with lack of strict sequence conservation of these residues among NLRP1 from different species, and with Y1445 as the most conserved residue at the dimerization interface (Supplementary Fig. 3c). Thus, while the UPA is required for UPA-CARD inflammasome signaling, the dimer likely promotes assembly to a much lesser degree.

If the NLRP1$^{CARD}$ dimerizes in the context of NLRP1-CT, why does the CARD8-CT fail to induce CARD8$^{CARD}$ dimerization? To address this question, we inspected the crystal packing interactions in the previously determined MBP-fused NLRP1$^{CARD}$ (PDB ID: 4IFP) and MBP-fused CARD8$^{CARD}$ (PDB ID: 4IKM) structures to see if dimers were observed[42,43]. In these preparations, the rigid MBP fusion kept NLRP1$^{CARD}$ and CARD8$^{CARD}$ from forming filaments. For NLRP1, all three independent molecules in the crystallographic asymmetric unit form a symmetrical dimer in the crystal lattice, which superimposes well with the dimer observed in the filament (Fig. 4e). These analyses suggest that NLRP1$^{CARD}$ has an intrinsic propensity to form dimers. In contrast, no CARD8$^{CARD}$ dimer was observed in its crystal lattice, and the corresponding CARD8$^{CARD}$ residues are not conserved across different species (Supplementary Fig. 3c). However, since CARD8$^{CARD}$ does not dimerize and the NLRP1$^{CARD}$ dimer interface plays limited functional role, we cannot exclude the possibility that the NLRP1$^{CARD}$ dimer results from the high protein concentrations used for structure determination. Regardless, multivalency drives CARD clustering to facilitate UPA-CARD filament formation and downstream signaling.

**The ASC type b surface nucleates caspase-1 polymerization**. We next investigated how heterooligomeric CARD-CARD interactions nucleate caspase-1 filament assembly to facilitate downstream signaling. We previously demonstrated unidirectional polymerization of ASC-nucleated caspase-1 filaments by nanogold labeling of ASC[36]. We further predicted that ASC$^{CARD}$ uses its type Ib, IIb, and IIIb surfaces to interact with the caspase-1$^{CARD}$ type Ia, IIa, and IIIa surfaces, because these interactions bury a larger total surface area than that of the reverse interactions between the type a surfaces of ASC and the type b surfaces of caspase-1[36]. To test this prediction, we designed mutations based on the ASC and caspase-1 filament structures to prevent filament elongation and thus observe a minimal unit of contact between ASC and caspase-1[7,36]. Specifically, we designed the type IIa W169G mutant of ASC$^{CARD}$ and the type IIb G20K mutant of caspase-1$^{CARD}$ and connected them in one polypeptide chain using a 5× GSS linker (Fig. 5a). As expected, these mutations did not impair the formation of an ASC$^{CARD}$–caspase-1$^{CARD}$ oligomer (Supplementary Fig. 4a), and multi-angle light scattering (MALS) measurement revealed a main peak of molecular mass of 86.3 kDa, corresponding to a tetramer of the ASC$^{CARD}$–caspase-1$^{CARD}$ fusion protein, or an octamer of CARDs with 4 ASC$^{CARD}$ molecules and 4 caspase-1$^{CARD}$ molecules (Fig. 5b).

We collected a cryo-EM dataset and found that despite its small size, the octamer 2D classes were quite detailed (Supplementary Fig. 4c). We then determined the 3D structure of the octamer at a resolution of 3.9 Å calculated by FSC between half maps (Supplementary Fig. 4d–f). There were eight CARD subunits in the map, four of which are similar to one another

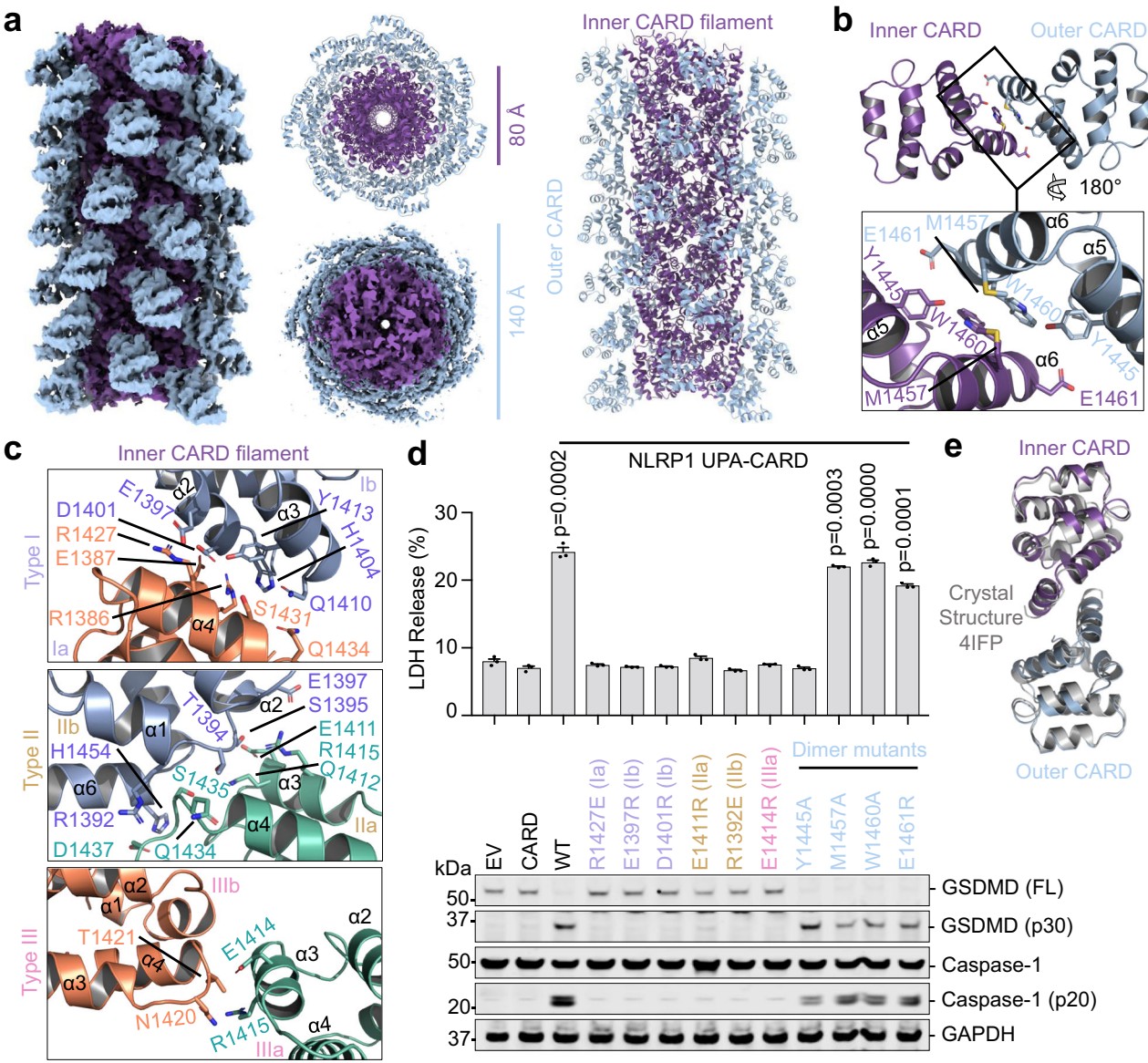

**Fig. 4 Cryo-EM structure of the dimeric NLRP1 CARD filament. a** Overall structure of the NLRP1-CT inflammasome. Each subunit of the inner helical CARD filament (purple) dimerizes with an outer CARD (blue). **b** Dimeric interface between an inner (purple) and outer (blue) CARD pair. **c** Zoom-ins of type I, II, and III helical interfaces between inner CARD molecules. **d** Structure-guided mutations of helical interfaces and the CARD–CARD dimer interface on UPA-CARD inflammasome signaling. LDH release (top) and western blot (bottom) are shown. *P*-values are compared to empty vector (EV) by two-sided Student's *t*-test. Exact *P*-values are provided in the Source Data File. Data are means ± SEM of *n* = 3 biological replicates. In **c**, **d** residues and mutations are labeled in the colors of the interface types as defined in Fig. 2c. **e** Alignment between Inner-outer CARD (purple and blue) and a dimer within the crystallographic symmetry unit of the NLRP1 CARD crystal structure (gray).

but differ from the remaining four, suggesting that there are four ASC<sup>CARD</sup> and four caspase-1<sup>CARD</sup> subunits in the map. The ASC<sup>CARD</sup> and caspase-1<sup>CARD</sup> structures were clearly distinguishable by their fit into the densities; in particular, the shorter α6 in caspase-1<sup>CARD</sup> and the longer α6 in ASC<sup>CARD</sup> allowed unambiguous placement of ASC<sup>CARD</sup> and capase-1<sup>CARD</sup> subunits (Fig. 5c, d and Supplementary Fig. 4g). In this assembly, there is one type I interaction and three type III interactions within ASC subunits and within caspase-1 subunits, and there are three type I interactions, four type II interactions, and one type III interaction between ASC and caspase-1 (Fig. 5e), which must have represented the optimal mode of interaction between ASC and caspase-1. In all three types of interactions between ASC and caspase-1, ASC always uses the type b surfaces (Ib, IIb, and IIIb) to interact with the type a surfaces (Ia, IIa, and IIIa) of caspase-1

(Fig. 5e). Of note, this arrangement requires a tetramer layer of ASC<sup>CARD</sup> to interact with a tetramer layer of caspase-1<sup>CARD</sup>, suggesting a hierarchical, rather than mixed, assembly of these death domains (Fig. 5f).

Extensive type I and II interfaces dominate the ASC<sup>CARD</sup> −caspase-1<sup>CARD</sup> assembly, with calculated buried surface areas of ~450, 600, and 220 Å² per partner for type I, II, and III interfaces, respectively. Detailed inspection of these interfaces revealed several clear interactions between ASC<sup>CARD</sup> and caspase-1<sup>CARD</sup>. These included E130 of ASC with R55 of caspase-1, W131 of ASC with R15 of caspase-1, and Q147 of ASC with D59 of caspase-1 at the type I interface, N128 of ASC with E37/E38 of caspase-1, S184 of ASC with N36 of caspase-1, and Q185/S186 of ASC with R33 of caspase-1 at the type II interface, T154 of ASC with R45 of caspase-1 and P153 of ASC with K42 with at the type III interface

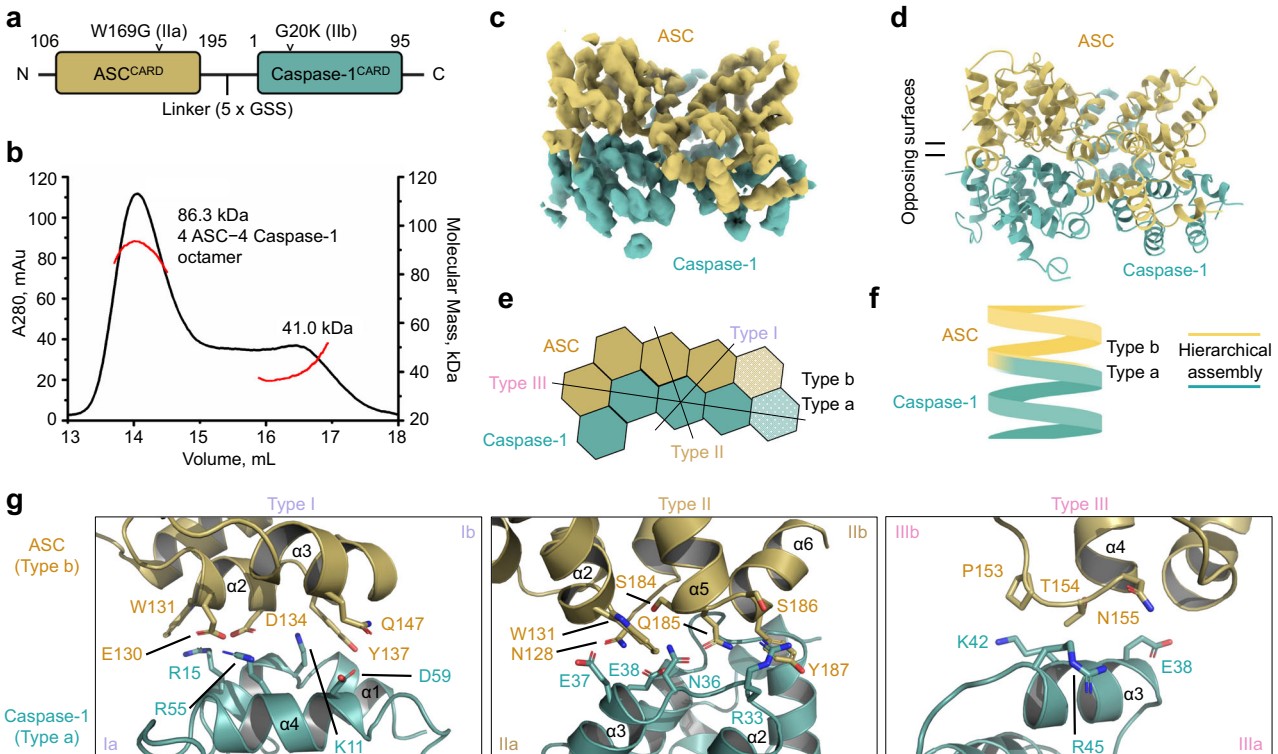

**Fig. 5 Purification and structure determination of an ASC^CARD–caspase-1^CARD octamer. a** Construct design of ASC^CARD (brown) and caspase-1^CARD (turquoise) linked by 5× GSS linker. **b** MALS data for the ASC^CARD–caspase-1^CARD complex, with a molecular mass of 86.3 kDa corresponding to a complex formed by four subunits of ASC^CARD and four subunits of caspase-1^CARD. **c** Cryo-EM density of the ASC^CARD–caspase-1^CARD octamer at 3.9 Å resolution in which a layer of ASC^CARD (gold) is located on top of a caspase-1^CARD layer (green). **d** Model of the ASC^CARD–aspase-1^CARD octamer. **e** Schematic diagram of the octamer complex with depicted interfaces. There are three type I interactions, four type II interactions, and one type III interaction between ASC and caspase-1. **f** Simplified illustration of ASC and caspase-1 hierarchy in inflammasome signaling. **g** Zoom-ins of type I, II, and III helical interfaces between ASC and caspase-1 molecules. In **c–g**, ASC^CARD and caspase-1^CARD are colored as defined in **c**.

(Fig. 5g). Indeed, a previous study demonstrated that K42E and D59R mutations in caspase-1^CARD, in addition to the E130R mutation in ASC^CARD, resulted in loss of ASC–caspase-1 interaction, supporting our assignment of these interaction surfaces[44].

**Predicted modes of interactions between NLRP1^CARD and ASC^CARD and between CARD8^CARD and caspase-1^CARD.** While most CARD-containing inflammasome proteins, including human NLRP1 and NLRC4, amplify signaling through the adapter protein ASC[32,34], CARD8 is the only known inflammasome sensor that cannot engage ASC and instead exclusively binds caspase-1[32]. NLRP1, conversely, exclusively interacts with ASC[32]. These differences in downstream signaling components could lead to different biological outcomes, given the signal amplification afforded by ASC and its differential expression across cell types.

The structures of CARD8^CARD and NLRP1^CARD filaments determined in this study allowed us to predict the modes of their CARD–CARD interactions, assuming a unidirectional assembly that has been elucidated for several other death domain superfamily members[3,36,45]. We first investigated the cross-sectional surface charges of CARD8, NLRP1, ASC, and caspase-1 filaments by extracting a layer of tetramer as defined in the ASC^CARD–caspase-1^CARD octamer from their respective filament structures. We then generated a CARD8^CARD–caspase-1^CARD hypothetical complex by fitting a layer of capsase-1^CARD tetramer to the CARD8^CARD filament structure by rigid-body fitting[46]. Of note, the interfacial side chains of CARD8^CARD and caspase-

1^CARD were not adjusted or energy minimized, therefore these models should be taken as highly suggestive. In the model in which the type b surfaces of the CARD8^CARD layer interact with the type a surfaces of the caspase-1^CARD layer, the charge complementarity between CARD8^CARD and caspase-1^CARD is apparent, especially with strong negative (CARD8) and positive (caspase-1) patches both mainly at the outer rim of the filament cross-sections (Fig. 6a). By contrast, in the opposing model wherein type a surfaces of the CARD8^CARD layer interact with the type b surfaces of the caspase-1^CARD layer, both interfaces display largely negative charges and should repel each other (Fig. 6b). Similarly, if the type a ASC^CARD layer is placed below the type b NLRP1^CARD layer, there is obvious charge complementarity with negative (NLRP1) and positive (ASC) patches both mainly near the center of the filament cross-sections (Fig. 6c). The reverse arrangement of ASC^CARD above NLRP1^CARD reveals repelling positive charges in the cross-sections of both ASC^CARD and NLRP1^CARD (Fig. 6d).

To further investigate the structural basis for CARD8^CARD-caspase-1^CARD and NLRP1^CARD–ASC^CARD interaction, we analysed the modeled interfaces type by type (Fig. 6e, f). Consistent with the surface charge analysis, the modeled CARD8^CARD (type b)–caspase-1^CARD (type a) interfaces are dominated by charged pairs including K469 of CARD8 to D52 of caspase-1 and D473 of CARD8 to R55 of caspase-1 at the type I interface, E523 of CARD8 to R33 of caspase-1, D467 and K469 of CARD8 to E38 and K42 of caspase-1, and a cluster of interactions involving Y527 and Y531 of CARD8 and K64 and Q67 of caspase-1 at the type II interface, and K493–S497 of CARD8 to K37, E38, K42, and R45 of caspase-1 at the type III interface (Fig. 5e).

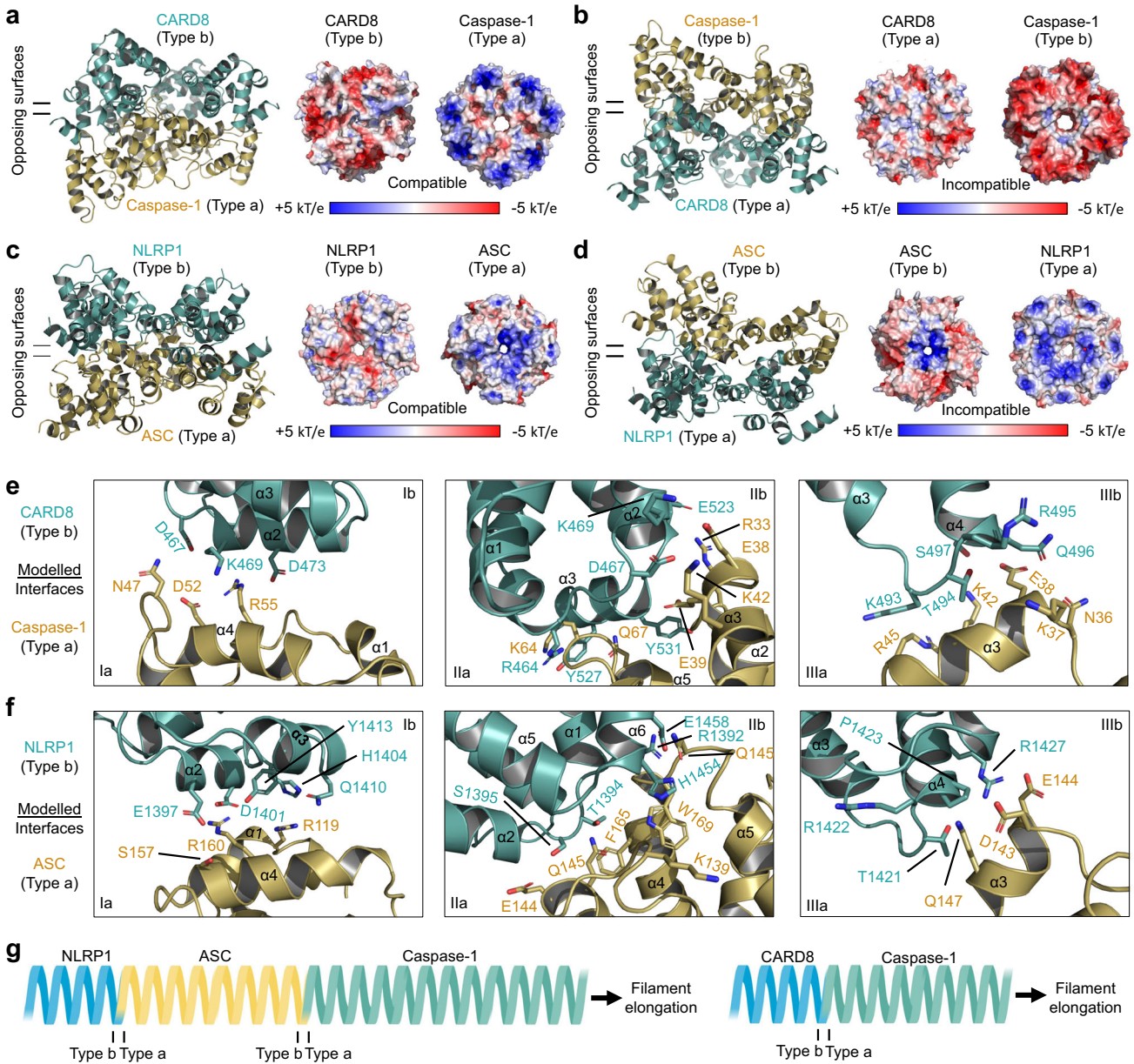

**Fig. 6 Modeled specific interfaces of the CARD8-CT and NLRP1-CT filaments for caspase-1 and ASC, respectively. a, b** Opposing arrangements of CARD8 and caspase-1 in their hypothetical modes of interaction. The electrostatic surfaces indicate that the negatively charged Type b surface of CARD8 most likely matches the positively charged Type a surface of caspase-1 in **a**. **c, d** Opposing arrangements of NLRP1 and ASC in their hypothetical modes of interaction. The electrostatic surfaces indicate that the largely negatively charged Type b surface of NLRP1 most likely matches the positively charged Type a surface of ASC in **c**. **e** Detailed modeled CARD-CARD type I–III interactions between CARD8 (green) and caspase-1 (gold). **f** Detailed modeled CARD-CARD type I–III interactions between NLRP1 (green) and ASC (gold). **g** Simplified illustration of NLRP1 and CARD8 hierarchy in inflammasome signaling. NLRP1 recruits ASC, followed by capase-1 recruitment. In contrast, CARD8 can only recruit caspase-1 directly. In **a–f**, the inflammasome sensors NLRP1 and CARD8 are colored turquoise while the downstream molecules ASC and caspase-1 are colored brown.

The NLRP1$^{CARD}$ (type b)−ASC$^{CARD}$ (type a) interfaces are dominated by both charged pairs and hydrophilic pairs including E1397 and D1401 of NLRP1 with R119 and R160 of ASC at the type I interface, T1394 and S1395 of NLRP1 with E144 and Q145 of ASC at the type II interface, and R1427 of NLRP1 with D143 and E144 of ASC at the type III interface (Fig. 6f). Thus, the structural analysis confirmed favorable interactions between the type b interfaces of NLRP1 and CARD8 with the type a interfaces of ASC and caspase-1, respectively (Fig. 6g). By contrast, ASC uses its type b surface to recruit caspase-1, suggesting a hierarchical inflammasome formation that proceeds through an ordered manner from NLRP1, to ASC, then to caspase-1, with

potential signal amplification (Fig. 6g). This ordered assembly is made possible by the use of two opposing ASC filament surfaces with specificity towards either NLRP1 or caspase-1.

In addition to the hierarchical NLRP1$^{CARD}$−ASC$^{CARD}$−caspase-1$^{CARD}$ assembly, oligomerized ASC$^{CARD}$ could, in principle, also nucleate the associated ASC$^{PYD}$ to form helical filaments, which in turn recruit more ASC through PYD–PYD interactions and allow the recruited ASC to template more caspase-1$^{CARD}$ filament formation through CARD–CARD interactions to further amplify the signal. In either scenario, however, these structural data suggest that ASC$^{CARD}$ uses opposing surfaces for interaction with NLRP1$^{CARD}$ and caspase-1$^{CARD}$.

## Discussion

The death-fold domain superfamily is widely represented in innate immune signaling pathways and mediates homo-oligomerization and hetero-oligomerization through filament formation[4,47,48]. In this study, we showed that NLRP1-CT and CARD8-CT also use their CARDs to assemble filamentous structures for inflammasome signaling. We posit that these CARD filaments are decorated by flexibly linked UPA sub-domains that are prone to oligomerization, and this additional multivalency reduces the threshold for CARD filament assembly (Fig. 3h). During the preparation of this manuscript, a complementary preprint on the structures of CARD8[CARD] and NLRP1[CARD] filaments was released[49]. Importantly, our studies together bolster the conclusion that the UPA subdomain on NLRP1-CT and CARD8-CT facilitates inflammasome assembly and signaling.

The role of the UPA subdomain in UPA-CARD inflammasomes might be analogous to other oligomerization domains in inflammasome sensors. For example, in the activated NLRC4 inflammasome, NBDs, and LRRs mediate its oligomerization into a disk-shaped signaling platform, clustering the attached CARDs towards the center of the assembly. The clustered CARDs then template the oligomerization of ASC and/or caspase-1 to stimulate inflammasome signaling[5,6,36,50]. NLRP3, which possesses a similar domain architecture, might also use a similar mechanism, as mutations in predicted NBD and LRR contacts abolished its inflammasome activity[30]. It was also recently reported that the isolated PYD of NLRP9 was not sufficient for inflammasome filament formation, implicating its other domains (LRR, NBD) in driving CARD oligomerization[51,52].

Other immune sensing pathways also employ CARD filament signaling strategies. In the RIG-I RNA sensing pathway, oligomerization of the RIG-I helicase domain on RNA clusters flexibly linked RIG-I 2CARDs, inducing their tetramerization. This RIG-I 2CARD tetramer, which represents one full helical turn, nucleates CARD filament formation of the MAVS adapter to stimulate interferon signaling[37,53,54]. In the CARMA1/BCL10/MALT1 (CBM) signalosome, CARMA1[CARD] nucleates the formation of a BCL10 filament through CARD–CARD interactions. However, the presence of an additional CARMA1 domain called the coiled coil region, which stimulates CARMA1 self-oligomerization, enhances BCL10 filament formation[38,55]. The role of the RIG-I helicase domain and the CARMA1 coiled coil region are thus similar to that of the UPA subdomain, by which self-oligomerization domains cluster flexibly linked CARDs to stimulate filament formation and signaling complex assembly.

Nucleated filament growth underlies the assembly and function of inflammasomes[3]. Such behavior allows signal amplification from a low concentration of a pathogen-activated or danger-activated sensor that acts as a nucleator to a robust host response[47]. Our data are consistent with this model, whereby CARD8[CARD] directly recruits and polymerizes caspase-1[CARD] through the favorable side of the growing filament seed. NLRP1[CARD], however, must recruit the adapter ASC to facilitate caspase-1 interaction and polymerization[32]. The specificity of the ASC[CARD] filament for NLRP1[CARD] on one side and caspase-1[CARD] on the other makes it a perfect bridge between the NLRP1 sensor and the caspase-1 effector, and polymerization of the ASC[PYD] might provide additional signal amplification. Thus, the ability to oligomerize and the specificity of the hetero-oligomerization determine the composition of particular signaling pathways and their biological outcomes.

## Methods

**Cloning, protein expression, and purification.** For structure determination, the UPA-CARD fragments of CARD8 (S297-L537, T60 isoform, Uniprot ID: Q9Y2G2-

5) and NLRP1 (S1213-S1473, isoform 1, Uniprot ID: Q9C000-1) were cloned into a modified pDB-His-MBP vector with an HRV 3C protease linker (His-MBP-3C-UPA-CARD). For in vitro filament formation experiments, the CARDs of CARD8 (P446-L537) and NLRP1 (D1374-S1473) were also cloned into the modified pDB-His-MBP vector. For in vitro analyses, expression constructs encoding CARD8-[CARD] and NLRP1[CARD] had an additional A452C (α1) or G1467C (α6) mutation, respectively, for cysteine labeling applications. The NLRP1 UPA (S1213–P1364) was also cloned into the modified pDB-His-MBP vector with an HRV 3C protease linker (His-MBP-3C-UPA). To generate the CARD8 His-MBP-3C-UPA (S297-P446) construct, a stop codon was introduced in the above CARD8 His-MBP-3C-UPA-CARD plasmid with Q5 mutagenesis. For cellular studies, the NLRP1-CT was cloned into pcDNA3.1 LIC 6A with a C-terminal FLAG tag (UPA-CARD-FLAG) and the CARD8-CT was cloned pcDNA3.1 LIC 6B encoding a C-terminal mCherry tag (UPA-CARD-mCherry). cDNA encoding ASC[CARD] (106–195) and caspase-1[CARD] (1–95) connected by a 5XGSS linker was cloned into the pDB-His-MBP vector with a tobacco etch virus (TEV)-cleavable N-terminal His₆-MBP tag using NdeI and Not I restriction sites. Mutations of W169G on ASC[CARD] (Type IIa) G20K on caspase-1[CARD] (Type IIb), and site-directed mutagenesis for all other relevant constructs were performed by QuikChange Mutagenesis (Agilent Technologies) or by Q5 mutagenesis (NEB). All plasmids were confirmed by Sanger sequencing. All constructs have been deposited to Addgene (https://www.addgene.org/Hao_Wu/). Study primers are listed in Supplementary Table 2.

To generate NLRP1-CT filaments, constructs were transformed into BL21 (DE3) cells, grown to OD600 0.6–0.8, cold shocked on ice water for 20 min, and induced overnight with 0.4 mM Isopropyl β-D-1-thiogalactopyranoside (IPTG) at 18 °C. The cells were harvested by centrifugation (4000 × g, 20 min) and lysed by sonication (2 s on 2 s off, 4 min total on, 55% power, Branson) in lysis buffer (25 mM Tris-HCl pH 7.5, 150 mM NaCl, 1 mM TCEP, SIGMAFAST protease inhibitor). Cell lysate was then centrifuged (40,000 × g, 1 h) and the soluble proteome was incubated with pre-equilibrated amylose resin for 2 h at 4 °C. Bound resin was then washed by gravity flow with 50 column volumes (CV) lysis buffer and eluted with 3 CV elution buffer (25 mM Tris-HCl pH 7.5, 150 mM NaCl, 1 mM TCEP, 50 mM maltose). Eluate from amylose resin was concentrated and purified as a range of small oligomers on a Superose 6 size exclusion column (Cytiva) in SEC buffer (25 mM Tris-HCl pH 7.5, 150 mM NaCl, 1 mM TCEP), concentrated again (500 μl, $A_{280} = 6$) using a 50 kDa spin column (Millipore), and cleaved overnight with MBP-3C protease (50 μl, $A_{280} = 3$) on ice. These cleavage conditions were optimized extensively to produce elongated filaments, as aggregation and incorporation of uncleaved MBP-tagged protein in the NLRP1-CT filaments limited further proteolysis of the MBP-tag and filament elongation. Despite extensive optimization, NLRP1-CT filaments still incorporated some uncleaved proteins containing MBP (Supplementary Fig. 2a). Filaments were further purified by size exclusion chromatography on a Superose 6 column (Cytiva) in lysis buffer and recovered from the column void fraction at a suitable concentration for cryo-EM ($A_{280} = 0.4$). UPA peptide coverage was confirmed by Mass Spectrometry at the HMS Taplin Mass Spectrometry Facility (Supplementary Fig. 2b).

To express and purify CARD8-CT filaments, expression and purification protocol followed as above until after elution from maltose resin. Eluate was concentrated (500 μL, $A_{280} = 6$) using a 50 kDa spin column (Millipore) and cleaved with MBP-3C protease (50 μL, $A_{280} = 3$) for 3 h at 37 °C to produce elongated filaments. Despite extensive optimization of these cleavage conditions, CARD8-CT filaments still incorporated some uncleaved proteins containing MBP (Supplementary Fig. 1a). Filaments were further purified by size exclusion chromatography on a Superose 6 column (Cytiva) in SEC buffer, recovered from the column void fraction, and slowly concentrated using a 0.5 mL spin concentrator (Millipore, 100 kDa MW cutoff). UPA peptide coverage was confirmed by Mass Spectrometry at the HMS Taplin Mass Spectrometry Facility (Supplementary Fig. 1b).

To obtain monomeric MBP-CARD proteins or oligomeric MBP-UPA proteins, constructs were expressed and purified similarly to the UPA-CARD filaments. Following elution from amylose resin, monomeric MBP-CARD proteins were concentrated to 0.5 mL (Amicon Ultra, 50 kDa MW cutoff) and further purified by size exclusion chromatography on a Superdex 200 column (Cytiva) in SEC buffer. Oligomeric MBP-UPA proteins were concentrated to 0.5 mL (Amicon Ultra, 50 kDa MW cutoff) and further purified by size exclusion chromatography on a Superose 6 column (Cytiva) in SEC buffer. Oligomeric MBP-UPA fractions were concentrated to 0.5 mL (Amicon Ultra, 50 kDa MW cutoff), followed by MBP cleavage by MBP-3C protease (50 μL, $A_{280} = 3$) for 3 h at RT and purified on another Superose 6 column (Cytiva) in SEC buffer. The peak fraction was taken for negative stain analysis.

Expression of ASC[CARD]-caspase-1[CARD] was similar to above. Cells were then collected and resuspended in lysis buffer (25 mM Tris-HCl at pH 8.0, 150 mM NaCl, 20 mM imidazole 5 mM β-ME), followed by sonication. The cell lysate was clarified by centrifugation (40,000 × g, 30 min) at 4 °C. The clarified supernatant containing the target protein was incubated with pre-equilibrated Ni-NTA resin (Qiagen) for 30 min at 4 °C. After incubation, the resin–supernatant mixture was poured into a gravity column and the resin was washed with 50 CV lysis buffer. The protein was eluted using lysis buffer supplemented with 500 mM imidazole. The Ni-NTA eluate was then incubated with TEV protease at 16 °C overnight, and the cleaved His₆-MBP tag was removed by passing the protein through an amylose resin column (Qiagen). The flow-through fraction containing the tag-free protein

was further purified using a Superdex 200 (10/300 GL) gel-filtration column (Cytiva, 25 mM Tris-HCl at pH 8.0, 150 mM NaCl, 2 mM DTT). Purified ASC$^{CARD}$-caspase-1$^{CARD}$ oligomer was assessed for absolute molecular mass by re-injecting the sample into a Superdex 200 column equilibrated with 25 mM Tris-HCl at pH 8.0, 150 mM NaCl, and 2 mM DTT. The chromatography system was coupled to a three-angle light scattering detector (mini-DAWN TRISTAR) and a refractive index detector (Optilab DSP) (Wyatt Technology). Data were collected every 0.5 s with a flow rate of 0.5 mL/min. Data analysis was carried out using ASTRA V.

**Negative stain EM and critical concentration determination.** Copper grids coated with layers of plastic and thin carbon film (Electron Microscopy Sciences) were glow discharged before 5 μL of purified proteins ($A_{280} = 0.1$) were applied. Samples were left on the grids for 1 min, blotted, and then stained with 1% uranyl formate for 30 s, blotted, and air dried. The grids were imaged on a JEOL 1200EX or Tecnai G$^2$ Spirit BioTWIN microscope at the Harvard Medical School (HMS) EM facility operating at 80 keV. For determining the critical concentration of CARD8$^{CARD}$ and NLRP1$^{CARD}$, we have purified the MBP fused protein and cleaved the MBP tag at different concentrations ranging from 1 to 30 μM, in order to engage filament formation. Samples were cleaved with 3C protease for 3 h at 37 °C prior to performing negatived stained EM imaging. Similarly, critical concentration of CARD8$^{UPA-CARD}$ and NLRP1$^{UPA-CARD}$ was determined by imaging cleaved protein samples with protein concentration ranging between 0.1 and 2 μM. The peak fractions of CARD8$^{UPA}$ and NLRP1$^{UPA}$ were taken for imaging.

**Cryo-EM data collection.** Cryo-EM data collection of NLRP1-CT filaments was conducted at the HMS Cryo-EM Center. Purified NLRP1-CT filaments ($A_{280} = 0.50$; 25 mM Tris-HCl pH 7.5, 150 mM NaCl, 1 mM TCEP) were loaded onto a glow-discharged C-flat grid (CF-1.2/1.3 400-mesh copper-supported holey carbon, Electron Microscopy Sciences), blotted for 4–5 s under 100% humidity at 4 °C, and plunged into liquid ethane using a Mark IV Vitrobot (ThermoFisher). Grids were screened for ice and particle quality prior to data collection. 1988 movies were acquired using a Titan Krios microscope (ThermoFisher) at an acceleration voltage of 300 keV equipped with a BioQuantum K3 Imaging Filter (slit width 25 eV), and a K3 direct electron detector (Gatan) operating in counting mode at 81,000× (1.06 Å pixel size). Automated data collection with SerialEM varied the defocus range between −0.8 and −2.2 μm with four holes collected per stage movement through image shift. All movies were exposed with a total dose of 52.3 e$^−$/Å$^2$ for 3.5 s fractionated over 50 frames.

Cryo-EM data collection of CARD8-CT filaments was conducted at the Pacific Northwest Center for Cryo-EM (PNCC). Purified CARD8-CT filaments ($A_{280} = 0.75$; 25 mM Tris-HCl pH 7.5, 150 mM NaCl, 1 mM TCEP) were loaded onto a glow-discharged Quantifoil grid (R1.2/1.3 400-mesh copper-supported holey carbon, Electron Microscopy Sciences), blotted for 4–5 s under 100% humidity at 4 °C, and plunged into liquid ethane using a Mark IV Vitrobot (ThermoFisher). Grids were screened for ice and particle quality prior to data collection. 1208 movies were acquired using a Titan Krios microscope (ThermoFisher) at an acceleration voltage of 300 keV equipped with a Falcon 3EC direct electron detector (ThermoFisher) operating in counting mode at 96,000× (0.8315 Å pixel size). Automated data collection with EPU varied the defocus range between −0.8 to −2.2 μm with three movies collected per hole through image shift prior to stage movement. All movies had an exposure time of approximately 40 s for a total dose of 40 e$^−$/Å$^2$ fractionated over 50 frames.

Cryo-EM data collection of the ASC-Caspase-1 CARD octamer was conducted at the HMS Cryo-EM Center. The CARD octamer sample ($A_{280} = 0.2$; 25 mM Tris-HCl pH 8.0, 150 mM NaCl, 2 mM DTT) was applied to a glow-discharged Quantifoil grid (R1.2/1.3 400-mesh gold-supported holey carbon, Electron Microscopy Sciences) blotted for 3–4 s under 100% humidity at 4 °C, and plunged into liquid ethane using a Mark IV Vitrobot (ThermoFisher). 5377 movies were acquired using a Titan Krios microscope (ThermoFisher) at an acceleration voltage of 300 keV equipped with a BioQuantum K3 Imaging Filter (slit width 20 eV), and a K3 direct electron detector (Gatan) operating in counting mode at 105,000× (0.825 Å pixel size). Automated data collection with SerialEM varied the defocus range between −0.8 to −2.5 μm with two shots collected in each of four holes per stage movement through image shift. All movies were exposed with a total dose of 57.12 e$^−$/Å$^2$ for 1.897 s fractionated over 50 frames.

**Cryo-EM data processing.** For the CARD8-CT filament, 1208 movies were aligned using RELION's implementation of the MotionCor2 algorithm[56]. The defocus values and contrast transfer functions (CTFs) of the motion-corrected micrographs were then computed and corrected for using CTTFFIND4.1[57]. 1181 micrographs were selected for further processing based on a maximum resolution criterion of 10 Å. Subsequently, start-end particle coordinates were manually picked and 287,568 particles were extracted in RELION[58], with a box size of 640 pixels, shift of 6 Å for each segment box, and two times binning[58,59] (320 pixel box). After 2D classification, 163,233 particles remained. An initial helical symmetry of 99.1° and 5.2 Å was derived from the power spectrum of the best 2D class average. An initial model was built with relion_helix_toolbox[58] using the initial helical parameters as input. 111,333 particles were selected after 3D classification,

which were re-extracted without binning and used for 3D refinement. Particle polishing and CTF refinement followed by final round of 3D refinement led to a refined helical symmetry −99.07° and 5.20 Å. Postprocessing in RELION led to a 3.3 Å reconstruction (Fig. 1 and Supplementary Fig. 1).

For the NLRP1-CT filament, movie frames were aligned using RELION's implementation of the MotionCor2 algorithm[56]. The defocus values and CTFs of the motion-corrected micrographs were then computed and corrected for using CTTFFIND4.1[57]. 1578 micrographs were selected for further processing based on a maximum CTF resolution criterion of 5 Å. Subsequently, start-end particle coordinates were manually picked, and 118,006 particles were extracted in RELION[58] with a box size of 512 pixels and shift of 48 for each segment box[58,59]. All particles were downscaled to a box size 360 to reduce processing time. After 2D classification, 55,083 particles remained. An initial helical symmetry of −100.8° and 5.1 Å was derived from the power spectrum of the best 2D class average. An initial model (helical lattice) was built with relion_helix_toolbox[58] using these initial helical parameters as input. 12,200 particles were selected after 3D classification and used for final 3D refinement, followed by CTF refinement and particle polishing prior to a final round of 3D refinement. The refined helical symmetry was −100.79° and 5.08 Å. Postprocessing in RELION led to a 3.6 Å reconstruction (Fig. 1 and Supplementary Fig. 2).

For the ASC$^{CARD}$-caspase-1$^{CARD}$ octamer, frames from 5377 movies were aligned using RELION's implementation of the MotionCor2 algorithm[56]. The defocus values and CTFs of the motion-corrected micrographs were then computed and corrected for using Gctf[60], and images were pre-screened with MicAssess[61]. 602,874 particles were picked in crYOLO[62], extracted with 1.65 Å pixel size (2× binning), and subjected to 2D and 3D classifications in RELION. 58,875 particles remained. A second round of particle picking was carried on in Gautomatch with selected 2D classes as template, and the previously selected 59,315 particles were excluded. 1,260,583 particles were picked and extracted with 1.65 Å pixel size. 61,147 remained after 2D and 3D classifications. These particles were combined with the previous particle stack to yield 120,462 total particles, followed by 3D refinement and particle polishing in RELION. 9409 particles were considered as duplicates and removed, based on a 60 Å minimum interparticle distance. Final 3D refinement and postprocessing with this set of 111,053 particles in RELION led to a 3.9 Å map. Map sharpening was performed with DeepEMhancer[63] using the "tightTarget" model (Fig. S5).

**Model building and display.** Model building was performed in program Coot[64]. The monomeric NLRP1$^{CARD}$ or CARD8$^{CARD}$ crystal structure (PDB ID: 4IFP[42] or 4IKM[43]) was fit to the NLRC4$^{CARD}$ filament model (PDB ID: 6N1I[36]) and used as initial models for refinement. Refinement was performed using Phenix[65] and Refmac[66]. For the ASC$^{CARD}$-caspase-1$^{CARD}$ octamer, subunit structures from ASC$^{CARD}$ (PDB ID: 6N1H[36]) and caspase-1$^{CARD}$ filaments (PDB ID: 5FNA[7]) were fitted individually into the octamer cryo-EM map, and refined using Phenix[65] and Refmac[66]. Structural representations were displayed and rendered using Pymol[67], UCSF ChimeraX[68], and UCSF Chimera[46].

**Molecular modeling.** For surface electrostatic analysis, CARD filaments of CARD8, NLRP1, caspase-1 (PDB ID: 5FNA[7]), and ASC (PDB ID: 6N1H[36]) were analysed with the Adaptive Poisson-Boltzmann Solver (APBS) in Pymol[67]. Electrostatics were scaled identically (±5) for comparison (Fig. 5a–d). To analyse the complementarity and the interaction between CARD8 and caspase-1, or between NLRP1 and ASC, a single layer of CARD8 or NLRP1 (four molecules) was extracted from its filament structure and fit to a single layer of caspase-1 or ASC. The UCSF Chimera "Matchmaker"[46] alignment tool was used to spread the RMSD resulting from differences in helical parameters among all four molecules. Interactions were analysed both below and above the fitted layer (a or b type).

**Cell death assay and immunoblotting.** HEK 293T cells stably expressing caspase-1 and GSDMD-V5 were seeded at $2 × 10^5$ cells/well in 12-well tissue culture dishes. The following day, cells were transfected with plasmids encoding for the indicated NLRP1 or CARD8 construct (0.02 μg), ASC for NLRP1 experiments (0.01 μg), and RFP (to 1 μg) using FuGENE HD according to manufacturer's instructions (Promega). Twenty-four hours later supernatants were analysed for LDH activity using the Pierce LDH Cytotoxicity Assay Kit (ThermoFisher) and lysates were evaluated by immunoblotting for caspase-1 (Cell Signaling Technology, #2225, 1:1000 dilution), GSDMD (Novus, 33422, 1:1000 dilution) and GAPDH (Cell Signaling Technology, 14C10, 1:1000 dilution) with secondary IRDye 800CW donkey anti-Rabbit (LI-COR, 926-32213, 1:10,000 dilution) imaged via Odyssey CLx (LI-COR). For CARD versus UPA-CARD titration experiments, cells seeded as described above were transfected with the indicated amount of CARD or UPA-CARD construct (20, 200, or 2000 ng), ASC for NLRP1 experiments (0.1 μg), and RFP (to 2.1 μg) per 125 μL optimum using FuGENE HD according to manufacturer's instructions (Promega). For NLRP1 dimer interface mutant titration experiments, cells seeded as described above were transfected with the indicated UPA-CARD construct (20, 2, or 0.2 ng), ASC (10 ng), and RFP (to 2 μg) per 125 μL optimum using FuGENE HD according to manufacturer's instructions (Promega). Supernatants were analysed for LDH release after 48 h using the Pierce LDH Cytotoxicity Assay Kit (ThermoFisher). Full uncropped gels are available in the accompanying Source Data file.

**Confocal imaging**. HEK 293T cultured in Dulbecco's modified Eagle's medium (DMEM) supplemented with 10% fetal bovine serum (FBS) were plated on CELLview 4-compartment dishes (Greiner Bio-One). HEK 293T cells were transfected with CARD8 WT and mutant UPA-CARD-mCherry constructs (0.02 µg) using lipofectamine 2000 (Thermo Fisher Scientific). Forty-eight hours after transfection, cells were fixed with 4% PFA for 10 min at RT. Cells were imaged using a spinning disk confocal Nikon microscope equipped with a Plan Apo 20×/1.3 air objective. Image analysis was performed in Fiji[69].

**Reporting summary**. Further information on research design is available in the Nature Research Reporting Summary linked to this article.

## Data availability

Raw cryo-EM data have been deposited in EMPIAR under the accession numbers EMPIAR-10567 (CARD8-CT filament), EMPIAR-10564 (NLRP1-CT filament), and EMPIAR-10566 (ASC$^{CARD}$-caspase-1$^{CARD}$ octamer). The cryo-EM structures have been deposited in the Electron Microscopy Data Bank (EMDB) with accession numbers EMD-22219 (CARD8-CT filament), EMD-22220 (NLRP1-CT filament), and EMDB-22233 (ASC$^{CARD}$-caspase-1$^{CARD}$ octamer). The atomic coordinates have been deposited in the protein databank (PDB) with accession numbers 6XKJ (CARD8-CT filament), 6XKK (NLRP1-CT filament), and 7KEU (ASC$^{CARD}$-caspase-1$^{CARD}$ octamer). Pymol/chimera session files are available on our Open Science Framework repository [https://doi.org/10.17605/OSF.IO/X7DV8]. Other data are available from the corresponding author upon reasonable request. Source data are provided with this paper.

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

## Acknowledgements

We thank Maria Ericsson at the HMS EM facility for advice and training, Zongli Li, Richard Walsh, Sarah Sterling, and Shaun Rawson at the Harvard Cryo-EM Center for Structural Biology for cryo-EM screening, data collection, training, and productive consultation, Paula Montero Llopis at the Microscopy Resources on the North Quad (MicRoN) core at HMS for microscope use, Ross Tomaino at the HMS Taplin Mass Spectrometry Facility, Theo Humphreys for cryo-EM screening and data collection at the Pacific Northwest Center for Cryo-EM (PNCC), Chen Xu and Kangkang Song for cryo-EM screening and data collection at UMass Medical School, BioRender for figure design, and SBGrid for computing support. This work was supported by National Institutes of Health grants DP1 HD087988 and R01 AI124491 to H.W., T32-GM007726 to L.R.H., the CRI Irvington Postdoctoral Fellowship to P.F., T32 GM007739-Andersen and F30 CA243444 to A.R.G., T32 GM115327-Tan to E.L.O., R01 AI137168 to D.A.B., and the Memorial Sloan Kettering Cancer Center Core Grant P30 CA008748 to D.A.B. A portion of this research was also supported by NIH grant U24GM129547 and performed at the PNCC at OHSU and accessed through EMSL (grid.436923.9), a DOE Office of Science User Facility sponsored by the Office of Biological and Environmental Research.

## Author contributions

H.W., L.R.H., and H.S. conceived the project idea and designed the study. L.R.H. designed constructs. L.R.H., H.S., P.F., and T.M.F. carried out preliminary filament expression and purification studies. L.R.H., P.F., and L.D. purified the filaments. J.R. designed and optimized the expression and purification of the ASC-caspase-1 octamer. T.M.F., L.R.H., L.D., and P.F. made cryo-EM grids for data collection. L.R.H. screened grids and collected cryo-EM data. Y.L., L.D., and L.R.H. analysed cryo-EM data. L.R.H. and L.D. performed model building and refinement. L.R.H. and L.D. designed and cloned mutants for in vitro and cell-based assays of the filaments. L.D. performed in vitro filament formation experiments and confocal microscopy. A.R.G. and E.L.O. performed cellular signalling experiments under the supervision of D.A.B. L.R.H. and H.W. wrote the manuscript with input from all authors.

## Competing interests

H.W. is a co-founder of Ventus Therapeutics. The other authors declare no competing interests.
