## [Peer Review File · Nature Communications]

REVIEWER COMMENTS

Reviewer #1 (Remarks to the Author):

NLRP1 and CARD8 are related cytosolic sensors that upon activation organize inflammasome signaling complexes to trigger caspase-1 activation, resulting in inflammatory cytokine maturation and/or pyroptotic cell death. NLRP1 and CARD8 bears constitutive autoproteolysis within their FIIND domain, resulting a non-covalent complex between N- and C-terminal fragments. Activation of NLRP1 and CARD8 requires both undergo functional N-terminal degradation, which unmask the active C-terminal fragment consisted of a CARD domain preceded with a UPA subdomain of FIIND. The molecular mechanism by which the active UPA-CARD fragments of NLRP1 and CARD8 assemble the inflammasome complexes and selectively recruit ASC or caspase-1 to mediate downstream signaling remains unresolved. In this manuscript, Hollingsworth and colleagues employ a combined structural and biochemical approach to investigate the structural mechanism governing NLRP1- and CARD8-mediated inflammasome assembly and signaling. They solved the cryo-EM structures of oligomeric UPA-CARD assembly of NLRP1 and CARD8, in which both CARDS form central helical filaments, resembling other known CARD filamentous structures, whereas the UPA is outside and flexibly linked to CARD filaments without an ordered organization. The CARDS of NLRP1 further forms a thicker filament by a unique dimerization interface. They also identify the structural basis in NLRP1 and CARD8 CARD filamentous assembly which enable NLRP1 and CARD8 to discriminate between ASC and pro-caspase-1 for diverse signalings. Further structural characterization of ASC-CARD and caspase-1-CARD interaction by an engineered CARD-CARD fusion protein provide more insights for understanding NLRP1-ASC-caspase-1 and CARD8-caspase-1 signaling. Overall, the experiments are well performed, and structural data are solid, which provides valuable insights into the molecular mechanism of NLRP1 and CARD8 inflammasome signaling, and has potential to be of broad interest in the innate immune field. However, there are still some gaps that need to be addressed in order to firmly establish the model proposed by the authors.

1) The authors only show circumstantial evidence in the cell-death assay that the UPA subdomain of FIIND is functionally important for NLRP1-ASC-caspase-1 signaling. They should provide more direct evidence to confirm that UPA upon functional release is intrinsically prone to oligomerization and facilitate the filamentous assembly of CARD.

2) To explore the structural basis for CARD8-CARD and NLRP1-CARD specificity for caspase-1-CARD and ASC-CARD respectively, the authors only modeled the potential interfaces between CARD8-CARD and caspase-1-CARD, or between NLRP1-CARD and ASC-CARD. They should also inspect the assumed interfaces between CARD8-CARD and ASC-CARD, or between NLRP1-CARD and caspase-1-CARD, to find out which residue substitutions render ASC not fit for CARD8 and caspase-1 not for NLRP1. Further mutagenesis validations to convince the interaction specificity are encouraged.

3) Is there any evidence to show that NLRP1-CARD and CARD8-CARD filaments are assembled unidirectionally? This is the prerequisite that both CARD filaments utilize only one end surface to prime the assembly of ASC-CARD and CASP1-CARD, respectively.

4) Given that mutagenesis analyses do not fully support the idea that unique CARD dimerization observed in the reconstituted NLRP1-CARD filaments is functionally important. The author should tone down the physiological significance of NLRP1-CARD dimerization.

Reviewer #2 (Remarks to the Author):

Hollingsworth et al. determined the structures CARD filaments from key inflammasome components whose assemblies are promoted by the UPA domain. The authors then used biochemical and cellular assays to test their structures. The authors also presented a low-resolution structure of a hetero-CARD octamer. These are noble structures of high importance to

understand the structural mechanisms of inflammasomes.

Here are my major concerns:

1. Figures 3 and 5 are not results based on "hard data." Although electrostatic complementarity analyses provide some general ideas about binding surfaces of macromolecules, they remain highly suggestive. These figures should be moved to the end of the paper (or supplementary figures) and be "discussed," instead of presented as "results." Also, in Fig. 3. The top and bottom electrostatics of ASC do not complement (the center of filament is blue in both directions), further weakening the use of electrostatics for drawing key conclusions.
2. The resolution of the ASC-caspase-1 octamer is too low to discern not only the identity of proteins, but also the directionality of their interactions (based on the gold-standard method, the resolution worse than $> 7\text{\AA}$). The large difference between 0.5 and 0.143 FSC indicates that the structure is also over-refined. The linker also might have forced the directionality of their interactions. Thus, the authors need to explicitly caution the readers that any conclusions from this structure should be taken as rather suggestive.
3. Together, the arguments/conclusions regarding the directionality and specificity of CARD-CARD interactions should all be moved to Discussion.

Minor:

1. Typo: Fig 2. Legend g. μm , not μM
2. It is interesting that the CARD8 filament shows a somewhat unique architecture, different from those of caspase-1 and ASC CARD filaments. Would that affect the specificity (directionality) and/or efficiency of its signaling activity?
3. The notes on the competing pre-print in Introduction should be moved to Discussion.
4. Can authors comment on the fact that filament formation is very slow and needs very high concentrations compared to in vivo scenarios? Please also list protein concentrations in negative stain EM experiments (e.g. Fig.2e).
5. Please fix the rotamer outliers in the CARD8 filament.
6. The outer CARD dimers are likely caused by high concentrations used for filament assembly and reconstruction.

Reviewer #3 (Remarks to the Author):

The authors present cryoEM structures of two filamentous oligomers of C-terminal portions of caspase recruitment domains, NLRP1 and Card8, as well as a ASC-caspase1 octamer. The association of the various domains, as well as their potential orientation relative to ASC and caspase1 binding partners, are discussed, although the UPA portion of both NLRP1 and Card8 are not evident.

While the work in this paper seems technically sound, the interpretation leaves something to be desired. The lack of any UPA signal in the structure makes any conclusions drawn about its function from these structures impossible, and so must rely on other biochemical evidence. This evidence (activity assays using LDH as a proxy for cell death, and caspase 1 levels) is indirect evidence of UPA function, and indeed there is very little to demonstrate that UPA is still present - though I do believe it may well be simply disordered. A western blot against UPA would demonstrate its presence in a purified fraction. The authors repeatedly discuss UPA dimerisation, for which I could find no supporting evidence of any sort presented in this manuscript.

Furthermore, the authors present conclusions about the binding interface between CARD and ASC

and Caspase-1 without giving any indication of the experimental procedures. The level of detail they ascribe to their analyses - namely amino acid level specificity - represents a drastic overreach of the evidence they present. This is particularly the case when they decide between model orientations "by visual inspection of the charge complementarity". Again, their analysis is plausible, but wholly unsupported by structural evidence or documented procedures. I strongly recommend aiming for a higher level of rigour in their analysis and presentation.

On the whole, I enjoyed the concept of the paper. In particular, I thought the octamer structure to be revealing in determining orientation of binding. The EM data seem adequate, and, at first glance, appear to be superior in resolution to the related structures cited in a recent bioRxiv deposition. The topic is of interest, and is pertinent to the field of inflammasome biology. The figures are beautifully done, and clear in what they attempt to convey. However, I find the interpretation of the data lacking in rigour, and overreaching in its scope. I do not recommend publication of this manuscript in its present form, but I encourage the authors either to introduce experiments (with experimental details) that support their plausible claims, or formulate an interpretation of the existing data consistent with the conclusions that can be drawn from the evidence at hand.

Concerns and comments:

Fig1 - The FSC plots in e) and h) are not at 1 at 10Å, even though you claim 3.5 and 3.7 Å resolution. This is not good. In general, the falloff of the curve in e) is strange, as the map-model correlation is better than the half-map correlation. How do you explain this?

- You seem to be using a 0.5 criterion for the map-model. A more conservative criterion seems to be warranted here given your FSC curves - 0.7 is used in some literature and may be more appropriate.

Page 5 - "One possible explanation is that CARD8UPA does not follow the CARD helical symmetry but is orderly associated with the central CARD8CARD filament." - CARD8CARD and UPA are covalently linked. How can UPA be "orderly associated", but not visible in the average? Wouldn't that be disordered? Have you tried focused classification instead of just subtraction and asymmetric classification? Also, your nomenclature is confusing. It sounds like you have two different filaments: "no UPA density was observed in the CARD8-CT filament structure, in which only the CARD8CARD filament was visible" You have one filament here: CARD8-CT. It's made of a CARD and UPA - one you can see, one you can't.

Fig 3 - It seems to me that the octamer in 3a) and the detailed interactions in 3c) are modeling exercises and are unsupported by structural data presented here. This is not very clear in the text or the figure caption. What is the evidence that ASC or Caspase-1 filaments will stack on top of CARD8? Nothing of the sort is presented.

Page 9 - "The functional role of UPA in inflammasome signalling is suggestive of its ability to dimerize or oligomerize such that its presence promotes CARD filament formation." - The data presented are not suggestive to me of any dimerisation of UPA (which is a feature also present in the schematic model in Fig4g). I see no structural or biochemical evidence presented here that corroborates the assertion of UPA dimerisation, though CARD dimerisation does seem to involve UPA presence.

Page 9 - "Based on these data and analyses, we propose a model of UPA- induced NLRP1CARD filament formation in which flexibly linked UPA dimers or oligomers promote the intrinsic tendency of NLRP1CARD dimerization and its helical polymerization to mediate inflammasome formation (Fig. 4g)." - And yet you don't see UPA in either structure... can you hypothesize why? Also, the orientation of CARD dimerisation would necessitate an asymmetry in the UPA domain on the N-Terminal end of your constructs. How would these differences be bridged in your model?

Page 10 - "To further elucidate the structural basis for the NLRP1-ASC interaction, we analysed the modelled interfaces type by type." - You do not describe how you arrive at this model. It does not seem to be based on structural data presented here, and, although plausible, remains speculative. You go on to give amino-acid level interactions for this model in the absence of structural data, or methods for how you generated it. This is not acceptable.

Page 11 - "and that UPA oligomerization decreases the threshold for filament assembly by locally

concentrating the CARDS." What is your evidence of this? I see a concentration-based test for CARD8, but conspicuously not for NLRP1 where the possibility of URA dimerisation is much higher. If this is true, shouldn't the filament assembly concentration be substantially lower?
Page 11 - "In NLRP1-CT, UPA dimerization or oligomerization is made apparent by an intrinsic propensity for CARD-CARD dimerization along an interface that does not participate in classical type I, II, or III helical interactions" - again, you are conflating CARD dimerisation with UPA dimerisation.

minor points:

Why no line numbers? Boooo to all of you!

Abstract: "oligomerized" => oligomerisation?

Abstract - UPA, ASC, not defined.

Abstract "low-resolution 4 ASCCARD-4 caspase-1CARD octamer" - unclear. Do you mean 4 Å resolution? Or 4 of each molecule? Rephrase.

Page 4 - ASC and UPA still not defined.

Page 5 - "By systematically optimizing cleavage conditions, we purified short (~100-200 nm) filaments that still contained some uncleaved MBP-tagged proteins." - I don't get it. You optimised cleavage conditions but still have MBP-tags in your filament? What were you optimising?

Fig1 - 1b) is great, and very helpful. Well done.

- Your 2D classes in 1d) are surrounded by a poorly-defined haze. Is this MBP?

Fig2d - the detail images are labeled Type I etc, but this label is not in a consistent location making it hard to find. the colour of this text is also not coordinated with the interaction type colour in the overview. Alternatively, the border of the detail could be coloured.

Fig 5 - "positive changes" => charges (twice)

Fig 5a - why "ASC or caspase-1" ? Which did you use? If the fit is the same, just say which you used and tell us the other fit is identical. If the point is only to show the orientation difference between a) and b), then be more explicit about that.

Page 11 - "hieratical" => hierarchical ?

Page 11 - "We posit that these CARD filaments are decorated by the flexibly linked UPA subdomains ..." Why are you positing this? Is there concern that UPA is not bound? Could it be cleaved? Do you have an antibody against the UPA region?

**We thank Reviewer 1 for their insights and suggestions.**

**Reviewer #1 (Remarks to the Author):**

NLRP1 and CARD8 are related cytosolic sensors that upon activation organize inflammasome
signaling complexes to trigger caspase-1 activation, resulting in inflammatory cytokine maturation
and/or pyroptotic cell death. NLRP1 and CARD8 bears constitutive autoproteolysis within their
FIIND domain, resulting a non-noncovalent complex between N- and C-terminal fragments.
Activation of NLRP1 and CARD8 requires both undergo functional N-terminal degradation, which
unmask the active C-terminal fragment consisted of a CARD domain preceded with a UPA
subdomain of FIIND. The molecular mechanism by which the active UPA-CARD fragments of
NLRP1 and CARD8 assemble the inflammasome complexes and selectively recruit ASC or
caspase-1 to mediate downstream signaling remains unresolved. In this manuscript,
Hollingsworth, David, Li, and colleagues employ a combined structural and biochemical approach
to investigate the structural mechanism governing NLRP1- and CARD8-mediated inflammasome
assembly and signaling.

They solved the cryo-EM structures of oligomeric UPA-CARD assembly of NLRP1 and
CARD8, in which both CARDS form central helical filaments, resembling other known CARD
filamentous structures, whereas the UPA is outside and flexibly linked to CARD filaments without
an ordered organization. The CARDS of NLRP1 further forms a thicker filament by a unique
dimerization interface. They also identify the structural basis in NLRP1 and CARD8 CARD
filamentous assembly which enable NLRP1 and CARD8 to discriminate between ASC and pro-
caspase-1 for diverse signalings. Further structural characterization of ASC-CARD and caspase-
1-CARD interaction by an engineered CARD-CARD fusion protein provide more insights for
understanding NLRP1-ASC-caspase-1 and CARD8-caspase-1 signaling. Overall, the
experiments are well performed, and structural data are solid, which provides valuable insights
into the molecular mechanism of NLRP1 and CARD8 inflammasome signaling, and has potential
to be of broad interest in the innate immune field.

**Responses:** We thank the reviewer for the positive comments.

However, there are still some gaps that need to be addressed in order to firmly establish the
model proposed by the authors.

**1. Comments:** The authors only show circumstantial evidence in the cell-death assay that the
UPA subdomain of FIIND is functionally important for NLRP1-ASC-caspase-1 signaling. They
should provide more direct evidence to confirm that UPA upon functional release is intrinsically
prone to oligomerization and facilitate the filamentous assembly of CARD.

**Responses:** The reviewer has identified a key lack of experimental evidence in our original
submission for which we planned to amend in the revision. To investigate whether the UPA
domain facilitates assembly of the UPA-CARD inflammasome, we titrated NLRP1 and CARD8
UPA-CARD versus their CARD-alone counterparts for filament formation *in vitro* and LDH release
in cells, and examined UPA alone for its aggregation property (new **Fig. 3** below). *In vitro*, 0.25
μM CARD8 or NLRP1 UPA-CARD (**Fig. 3b and d**) induced filament formation; by contrast, only
at 15 μM CARD8^{CARD} formed filaments (**Fig. 3a**) and even at 30 μM , NLRP1^{CARD} failed to form
filaments (**Fig. 3c**). In cells, at a lower amount of plasmids transfected (20 ng or 200
45 ng/transfection), the NLRP1 or CARD8 UPA-CARD induced more LDH release than their CARD
alone counterparts; only at the highest amount of plasmids transfected (2000 ng/transfection),
NLRP1 or CARD8 UPA-CARD and CARD induced similar levels of LDH release (**Fig. 3f and g**).
Finally, purified MBP-fused UPAs from NLRP1 and CARD8 were already oligomers even with the
large MBP tag that often prevents oligomerization (**Fig. 3e**). Thus, while not absolutely required,
the UPA subdomains lower the threshold needed for inflammasome signaling due to their ability

to oligomerize, which is particularly important in physiological contexts when the endogenous
 protein concentrations are low.

 **Figure 3 | The UPA subdomain reduces the concentration required for inflammasome**
 **formation.** (a-b) Concentration-dependent formation of CARD8^{CARD} (a) and CARD8^{UPA-CARD} (b)
 filaments, visualized by negative staining EM. Purified monomeric MBP-fused proteins were
 cleaved at concentrations ranging from 0.1-15 μ M. CARD8^{CARD} filaments did not appear
 consistently until 15 μ M. CARD8^{UPA-CARD} filaments formed consistently at 0.25 μ M. Arrows in (a)
 indicates single, unbundled filaments. (c-d) Concentration-dependent formation of NLRP1^{CARD}
 (c) and NLRP1^{UPA-CARD} (d) filaments. Purified monomeric MBP-fused proteins were cleaved at
 concentrations ranging from 0.1-30 μ M. NLRP1^{CARD} filaments did not appear even at 30 μ M
 concentration, whereas NLRP1^{UPA-CARD} filaments formed consistently at 0.25 μ M. (e) MBP-fused
 CARD8^{UPA} and NLRP1^{UPA} (25 nM) imaged by negative staining EM, which already showed
 oligomers even with the large MBP tag. (f) Titration of CARD8^{CARD} and CARD8^{UPA-CARD} in
 HEK293T cells stably expressing caspase-1 and GSDMD. A ~100-fold higher amount of the
 CARD8^{CARD} plasmid was needed to achieve a comparable level of cell death (marked by LDH
 release) by CARD8^{UPA-CARD}. (g) Titration of NLRP1^{CARD} and NLRP1^{UPA-CARD} with ASC co-
 expression in HEK293T cells stably expressing caspase-1 and GSDMD. A ~100-fold higher
 amount of the NLRP1^{CARD} plasmid was needed to achieve a comparable level of cell death
 (marked by LDH release) by NLRP1^{UPA-CARD}. (h) Model of UPA-enhanced inflammasome
 formation. The UPA subdomain increases the multivalency of the CARD8 and NLRP1
 inflammasomes, lowering the concentration required to nucleate filaments and subsequently
 signal.

Additionally, we recently released a pre-print of our human DPP9-NLRP1 complex structure
(Hollingsworth*, Sharif*, and Griswold* *et. al.*, BioRxiv,
<https://www.biorxiv.org/content/10.1101/2020.08.14.246132v2>), in which we observed a front-to-
back UPA-UPA oligomerization interface. This interface was also present on the rat DPP9-NLRP1
complex structure (Huang* and Zhang* *et. al.*, BioRxiv,
<https://www.biorxiv.org/content/10.1101/2020.08.13.250241v2>). Point mutations of this UPA-
UPA interface abolished NLRP1 inflammasome signaling in cells while maintaining
autoproteolytic functions of the protein, and impaired UPA-CARD filament formation. Together,
these results provide a strong premise for a bona fide functional role of UPA oligomerization in
NLRP1 and CARD8 inflammasomes.

**2. Comments:** To explore the structural basis for CARD8-CARD and NLRP1-CARD specificity
for caspase-1-CARD and ASC-CARD respectively, the authors only modeled the potential
interfaces between CARD8-CARD and caspase-1-CARD, or between NLRP1-CARD and ASC-
CARD. They should also inspect the assumed interfaces between CARD8-CARD and ASC-
CARD, or between NLRP1-CARD and caspase-1-CARD, to find out which residue substitutions
render ASC not fit for CARD8 and caspase-1 not for NLRP1. Further mutagenesis validations to
convince the interaction specificity are encouraged.

**Responses:** We thank the reviewer for the suggestion. We modeled only the potential interfaces
between CARD8-CARD and caspase-1-CARD, or between NLRP1-CARD and ASC-CARD
because the specificity of CARD8 for caspase-1 and NLRP1 for ASC was demonstrated
previously (Ball *et. al.*, Life Sci Alliance 2020). NLRP1 requires ASC for caspase-1-mediated cell
death, co-localizes with ASC, and pulls down ASC oligomers when crosslinked by DSS
(disuccinimidyl suberate). CARD8 does not require ASC, does not co-localize with ASC, and does
not pull-down ASC when DSS crosslinked. In a split luciferase assay, it was also evident that
CARD8^{CARD} exclusively binds caspase-1^{CARD}, whereas NLRP1^{CARD} exclusively engages ASC^{CARD}
(Ball *et. al.*, Life Sci Alliance 2020).

D. P. Ball *et al.*, Caspase-1 interdomain linker cleavage is required for pyroptosis. *Life Sci Alliance*
**3**, (2020).

We agree that all potential interfaces should be modeled extensively, and while not
presented, we had inspected every permutation of the potential CARD8/ASC/caspase-1 and
NLRP1/ASC/caspase-1 complexes (Figure for the models below). What we found is that all the
“impossible” interfaces have predicted steric and/or electrostatic clashes to justify why these
interactions do not occur. However, the CARD domains of ASC, caspase-1, CARD8 and NLRP1
are quite different in sequence despite having the same fold (**Table** below), with NLRP1 and ASC
being the most similar. Thus, many residues are responsible for this specificity (**Figure** for the
different residues in the three types of interfaces below) and it would be difficult to pin-point a
particular specificity determinant.

Figure | Structural analysis on the incompatibility of various hypothetical CARD-CARD interactions involving CARD8 and NLRP1, highlighting potential steric and electrostatic clashes (dotted red ovals). (a) Modelled CARD8-Caspase-1 interaction in the reverse direction show in Fig. 6e. (b-c) Modelled CARD8-ASC interactions in two opposite directions. (d) Modelled NLRP1-ASC interaction in the reverse direction show in Fig. 6f. (e-f) Modelled NLRP1-Caspase-1 interactions in two opposite directions.

**Table** | Pairwise identity between the CARDS of the following proteins based on structure
 alignment by DALI.

	Caspase-1	CARD8	ASC	NLRP1
Caspase-1		28.7%	18.2%	17.4%
CARD8	28.7%		26.1%	27.6%
ASC	18.2%	26.1%		44.8%
NLRP1	17.4%	27.6%	44.8%	

**Figure** | Structure-based sequence alignment computed with DALI, using ASC^{CARD} (PDB ID:
 6N1H), Caspase-1^{CARD} (PDB ID: 5FNA), CARD8^{CARD} (this paper), and NLRP1^{CARD} (this paper) as
 input. Lower case letters indicate residues that are not aligned by structure. The 6 interfaces in
 these proteins are highlighted, showing their similarities and differences.

**3. Comments:** Is there any evidence to show that NLRP1-CARD and CARD8-CARD filaments
 are assembled unidirectionally? This is the prerequisite that both CARD filaments utilize only one
 end surface to prime the assembly of ASC-CARD and CASP1-CARD, respectively.

**Responses:** We have not demonstrated the directionality of NLRP1-CARD and CARD8-CARD
 filaments here experimentally. While the UPA-CARD filaments of either CARD8 or NLRP1 are too
 short (~100 nm, **Fig. 3b, d**) so that they will only show up at “dots” rather than “filaments” under
 confocal microscopy due to the diffraction limit of light, fluorescently labeled CARD8^{CARD}
 polymerized too slowly that we have not been able to capture its filament formation under time
 lapse microscopy.

However, unidirectional polymerization for death-domain filaments has been demonstrated
 directly for several death-domain systems, including BCL10-CARD by time-lapse confocal
 microscopy (David et. al, 2018), immunogold end-labeling of both AIM2-PYD/ASC-PYD and
 NLRP3-PYD/ASC-PYD filaments (Lu*, Magupalli*, and Ruan* et. al., 2014), and end-labeling of
 ASC-CARD/Casp-1-CARD and NLRC4-CARD/Caspase-1-CARD filaments (Li et. al, 2019).
 Structure modeling and simulations of NLRP6-PYD (Shen et. al, 2019) also suggested
 unidirectional assembly. Thus, unidirectional signaling appears to be a paradigm for death-
 domain family filaments. In the case of NLRP1 and CARD8, their specificity for ASC and caspase-
 1, respectively, has been demonstrated previously (Ball et. al., Life Sci Alliance 2020), and we
 hope that the reviewer is satisfied with our structural analysis which helps to further narrow down
 which surface(s) of the NLRP1 and CARD8 filaments are responsible for the interactions based
 on modeled structural compatibility (**Fig. 6a, c**).

159 L. David et al., Assembly mechanism of the CARMA1-BCL10-MALT1-TRAF6 signalosome.
 *Proc Natl Acad Sci U S A* **115**, 1499-1504 (2018).

- 161 A. Lu *et al.*, Unified polymerization mechanism for the assembly of ASC-dependent
 inflammasomes. *Cell* **156**, 1193-1206 (2014).
 Y. Li *et al.*, Cryo-EM structures of ASC and NLRC4 CARD filaments reveal a unified mechanism
 of nucleation and activation of caspase-1. *Proc Natl Acad Sci U S A* **115**, 10845-10852
 (2018).
 C. Shen *et al.*, Molecular mechanism for NLRP6 inflammasome assembly and activation. *Proc*
 *Natl Acad Sci U S A* **116**, 2052-2057 (2019).
 D. P. Ball *et al.*, Caspase-1 interdomain linker cleavage is required for pyroptosis. *Life Sci*
 *Alliance* **3**, (2020).

**4. Comments:** Given that mutagenesis analyses do not fully support the idea that unique CARD
 dimerization observed in the reconstituted NLRP1-CARD filaments is functionally important. The
 author should tone down the physiological significance of NLRP1-CARD dimerization.

**Responses:** To further elucidate the role of the CARD dimer, we tried titrating the concentration
 of these dimer mutants in a cellular system and also investigated their ability to form filaments at
 a concentration of the UPA-CARD sufficient for the WT (**Supplementary Fig. 3a-b**). While the
 Y1445A mutant abolished filament formation, the other mutants resembled the wild-type UPA-
 CARD. Given this new data, the reviewer's suggestion and that the dimerization surface of NLRP1
 is only conserved in mammalian NLRP1, but not in more remote NLRP1 or any CARD8, we toned
 down discussion on the functional significance of NLRP1-CARD dimerization.

**Supplementary Figure 3 | The NLRP1 dimer interface has limited impact on inflammasome**
 **formation.** (a) LDH release when NLRP1 UPA-CARD dimer mutants was titrated in HEK293T
 cells stably expressing caspase-1 and GSDMD, with ASC co-expression. All mutants behaved
 similarly as the WT. (b) Filament formation by NLRP1 UPA-CARD dimer mutants *in vitro*. At the
 concentration used, most of these mutants formed filaments, like WT (Fig. 3d), except for the
 Y1445A mutant, which was defective in filament formation, consistent with its impairment in LDH
 release (Fig. 4d). (c) Multiple sequence alignment (Clustal Omega) between NLRP1^{CARD} and
 CARD8^{CARD} homologs, coloured with ESPrnt⁷³. The NLRP1 dimer interface residues (annotated

with purple asterisks) are largely conserved across NLRP1 paralogs in mammals, but not in
zebrafish NLRP1 or all CARD8 paralogs.

**We thank Reviewer 2 for their enthusiasm and well as their insights and suggestions.**

**Reviewer #2 (Remarks to the Author):**

Hollingsworth, David, and Li et al. determined the structures CARD filaments from key
inflammasome components whose assemblies are promoted by the UPA domain. The authors
then used biochemical and cellular assays to test their structures. The authors also presented a
low-resolution structure of a hetero-CARD octamer. These are noble structures of high
importance to understand the structural mechanisms of inflammasomes.

**Here are my major concerns:**

**1. Comments:** Figures 3 and 5 are not results based on “hard data.” Although electrostatic
complementarity analyses provide some general ideas about binding surfaces of macromolecules,
they remain highly suggestive. These figures should be moved to the end of the paper (or
supplementary figures) and be “discussed,” instead of presented as “results.” Also, in Fig. 3. The
top and bottom electrostatics of ASC do not complement (the center of filament is blue in both
directions), further weakening the use of electrostatics for drawing key conclusions.

**Responses:** We agree with the reviewer and thus now removed the most speculative parts of
Figs. 3 and 5, and combined them into the new **Fig. 6** (below) that we showed as the last section
in **Results**. Here we built on the known specificity of CARD8 for caspase-1 and NLRP1 for ASC
(Ball *et. al.*, Life Sci Alliance 2020), and used the filament surfaces of CARD8 and NLRP1 to
deduce which side of CARD8 recruits which side of caspase-1, and which side of NLRP1 recruits
which side of ASC. We also clarified that our modelling is suggestive rather than conclusive in the
text.

D. P. Ball *et al.*, Caspase-1 interdomain linker cleavage is required for pyroptosis. *Life Sci*
*Alliance* **3**, (2020).

Of note, although not directly shown for the CARD8 and the NLRP1 system, unidirectional
polymerization for death-domain filaments has been demonstrated directly for several death-
domain systems, including BCL10-CARD by time-lapse confocal microscopy (David *et. al.*, 2018),
immunogold end-labeling of both AIM2-PYD/ASC-PYD and NLRP3-PYD/ASC-PYD filaments
(Lu*, Magupalli*, and Ruan* *et. al.*, 2014), and end-labeling of ASC-CARD/Casp-1-CARD and
NLRC4-CARD/Caspase-1-CARD filaments (Li *et. al.*, 2019). Structure modeling and simulations
of NLRP6-PYD (Shen *et. al.*, 2019) also suggested unidirectional assembly. Thus, unidirectional
signaling appears to be a paradigm for death-domain family filaments.

233 L. David *et al.*, Assembly mechanism of the CARMA1-BCL10-MALT1-TRAF6 signalosome.
*Proc Natl Acad Sci U S A* **115**, 1499-1504 (2018).

235 A. Lu *et al.*, Unified polymerization mechanism for the assembly of ASC-dependent
inflammasomes. *Cell* **156**, 1193-1206 (2014).

Y. Li *et al.*, Cryo-EM structures of ASC and NLRC4 CARD filaments reveal a unified mechanism
of nucleation and activation of caspase-1. *Proc Natl Acad Sci U S A* **115**, 10845-10852
(2018).

C. Shen *et al.*, Molecular mechanism for NLRP6 inflammasome assembly and activation. *Proc*
*Natl Acad Sci U S A* **116**, 2052-2057 (2019).

Figure 6 | Modelled specific interfaces of the CARD8-CT and NLRP1-CT filaments for caspase-1 and ASC, respectively. (a-b) Opposing arrangements of CARD8 and caspase-1 in their hypothetical modes of interaction. The electrostatic surfaces indicate that negatively charged Type b surface of CARD8 most likely matches the positively charged Type a surface of caspase-1 in (a). (c-d) Opposing arrangements of NLRP1 and ASC in their hypothetical modes of interaction. The electrostatic surfaces indicate that largely negatively charged Type b surface of NLRP1 most likely matches the positively charged Type a surface of ASC in (c). (e) Detailed modelled CARD-CARD type I-III interactions between CARD8 (green) and caspase-1 (gold). (f) Detailed modelled CARD-CARD type I-III interactions between NLRP1 (green) and ASC (gold). (g) Simplified illustration of NLRP1 and CARD8 hierarchy in inflammasome signalling. NLRP1 recruits ASC, followed by capase-1 recruitment. In contrast, CARD8 can only recruit caspase-1 directly.

2. Comments: The resolution of the ASC-caspase-1 octamer is too low to discern not only the identity of proteins, but also the directionality of their interactions (based on the gold-standard method, the resolution worse than $> 7\text{\AA}$). The large difference between 0.5 and 0.143 FSC

indicates that the structure is also over-refined. The linker also might have forced the directionality
 of their interactions. Thus, the authors need to explicitly caution the readers that any conclusions
 from this structure should be taken as rather suggestive.

**Responses:** We apologize for including the map-model FSC when we initially only rigidly fit the
 known structures of caspase-1^{CARD} and ASC^{CARD} into the low-resolution density. While the gold-
 standard method indicated that the half-map correlation was 5.0 Å at FSC=0.143 (> 7 Å for the
 map-model correlation at FSC=0.5), some bulky side chains, and distinctive differences in the
 lengths of helix 6, allowed us to unambiguously determine the identity of the 8 chains. However,
 we realized that we could do better than this 1000-micrograph dataset collected on a K2-equipped
 200 keV Talos Arctica, and went ahead to collect a ~5500-micrograph new dataset on a K3-
 equipped 300 keV Titan Krios microscope with an energy filter. This new dataset facilitated the
 calculation of a map at 3.9 Å with many discernable side chains (**Fig. 5**, below), particularly some
 at protein-protein interfaces. The map-model FSC is now at a 4.1 Å resolution, much more similar
 to the resolution indicated by the map-map FSC. Below we also share the fitting of ASC^{CARD} and
 caspase-1^{CARD} into the density (**Supplementary Fig. 4**, below).

 **Figure 5 | Purification and structure determination of an ASC^{CARD}-caspase-1^{CARD} octamer.**
 (a) Construct design of ASC^{CARD} and caspase-1^{CARD} linked by 5 x GSS linker. (b) MALS data for
 the ASC^{CARD}-caspase-1^{CARD} complex, with a molecular mass of 86.3 kDa corresponding to a
 complex formed by 4 subunits of ASC^{CARD} and 4 subunits of caspase-1^{CARD}. (c) Cryo-EM density
 of the ASC^{CARD}-caspase-1^{CARD} octamer at 3.9 Å resolution in which a layer of ASC^{CARD} (gold) is
 located on top of a caspase-1^{CARD} layer (green). (d) Model of the ASC^{CARD}-caspase-1^{CARD}
 octamer. (e) Schematic diagram of the octamer complex with accurately depicted interfaces.
 There are three type I interactions, four type II interactions, and one type III interaction between
 ASC and caspase-1. (f) Simplified illustration of ASC and caspase-1 hierarchy in inflammasome
 signalling. (g) Zoom-ins of type I, II, and III helical interfaces between ASC and caspase-1
 molecules.

**Supplementary Figure 4 | Structure determination of the ASC-Caspase-1 octamer.** (a)
 Purification of the ASC-Caspase-1 octamer. The indicated fractions were run on the
 corresponding SDS-PAGE gel. (b) Representative cryo-EM micrograph. (c) Representative 2D
 class averages. (d) Data processing flow chart for structure determination in RELION. (e) Gold-
 standard FSC and map-model correlation plots of the ASC-Caspase-1 octamer, which gave an
 overall resolution of 3.9 Å. (f) Local resolution of the octamer, calculated with RELION. (g) Fittings
 of ASC^{CARD} and Caspase-1^{CARD} into cryo-EM densities. The incorrectness of the reverse fittings
 is apparent.

**3. Comments:** Together, the arguments/conclusions regarding the directionality and specificity
 of CARD-CARD interactions should all be moved to Discussion.

**Responses:** We have added introduction on the unidirectionality of death domain family filaments
 and removed most of the speculations (see also Responses to Comment 1).

**Minor:**

1. Typo: Fig 2. Legend g. μm , not μM

**Responses:** Addressed.

2. It is interesting that the CARD8 filament shows a somewhat unique architecture, different from
 those of caspase-1 and ASC CARD filaments. Would that affect the specificity (directionality)
 and/or efficiency of its signaling activity?

**Responses:** In our analysis, although the appearance of the CARD8 filament may be somewhat
different as shown by the “holy” surface on the filament, the underlying mechanism of assembly remains
similar to other death domain family filaments and possesses similar helical symmetry as other CARDS.
In this case, the CARD8^{CARD} filament cross-section (mainly negatively charged) is most compatible with
that of caspase-1^{CARD} (positively charged) (**Fig. 6a**). While ASC^{CARD} filament cross-section also has
positive charges, the positive charges are mainly clustered near the center (**Fig. 6c**); in contrast, the
negatively charges on CARD8^{CARD} filament cross-section are more towards the periphery (**Fig. 6a**).

**3. The notes on the competing pre-print in Introduction should be moved to Discussion.**

**Responses:** Addressed.

**4. Can authors comment on the fact that filament formation is very slow and needs very high**
**concentrations compared to in vivo scenarios? Please also list protein concentrations in negative**
**stain EM experiments (e.g. Fig.2e).**

**Responses:** We now list protein concentrations used in all negative staining EM experiments in
the figure legends. In our new data, we showed that filament formation for UPA-CARD occurs at
a much lower concentration than CARD alone (new **Fig. 3** below). *In vitro*, 0.25 μ M CARD8 or
NLRP1 UPA-CARD (panel b and d) induced filament formation; by contrast, only at 15 μ M CARD8
CARD formed filaments (panel a) and even at 30 μ M, NLRP1 CARD failed to form filaments (panel
331 d). In cells, at a lower amount of plasmids transfected (20 ng or 200 ng/transfection), the NLRP1
or CARD8 UPA-CARD induced more LDH release than their CARD alone constructs; only at the
highest amount of plasmids transfected (2000 ng/transfection), NLRP1 or CARD8 UPA-CARD
and CARD induced similar levels of LDH release (panel f and g). Thus, with a critical concentration
of around 0.25 μ M for UPA-CARD, the correspondence with the in vivo scenarios can now be
imagined, as many of these signaling proteins exist at a concentration of around 1 μ M in cells.

Additionally, we recently released a pre-print of our human DPP9-NLRP1 complex structure
(Hollingsworth*, Sharif*, and Griswold* *et. al.*, BioRxiv,
<https://www.biorxiv.org/content/10.1101/2020.08.14.246132v2>), in which we observed a front-to-
back UPA-UPA oligomerization interface. This interface was also present on the rat DPP9-NLRP1
complex structure (Huang* and Zhang* *et. al.*, BioRxiv,
<https://www.biorxiv.org/content/10.1101/2020.08.13.250241v2>). Point mutations of this UPA-
UPA interface abolished NLRP1 inflammasome signaling in cells while maintaining
autoproteolytic functions of the protein, and impaired UPA-CARD filament formation. Together,
these results provide a strong premise for a bona fide functional role of UPA oligomerization in
NLRP1 and CARD8 inflammasomes.

Figure 3 | The UPA subdomain reduces the concentration required for inflammasome formation. (a-b) Concentration-dependent formation of CARD8^{CARD} (a) and CARD8^{UPA-CARD} (b) filaments, visualized by negative staining EM. Purified monomeric MBP-fused proteins were cleaved at concentrations ranging from 0.1-15 μ M. CARD8^{CARD} filaments did not appear consistently until 15 μ M. CARD8^{UPA-CARD} filaments formed consistently at 0.25 μ M. Arrows in (a) indicates single, unbundled filaments. (c-d) Concentration-dependent formation of NLRP1^{CARD} (c) and NLRP1^{UPA-CARD} (d) filaments. Purified monomeric MBP-fused proteins were cleaved at concentrations ranging from 0.1-30 μ M. NLRP1^{CARD} filaments did not appear even at 30 μ M concentration, whereas NLRP1^{UPA-CARD} filaments formed consistently at 0.25 μ M. (e) MBP-fused CARD8^{UPA} and NLRP1^{UPA} (25 nM) imaged by negative staining EM, which already showed oligomers even with the large MBP tag. (f) Titration of CARD8^{CARD} and CARD8^{UPA-CARD} in HEK293T cells stably expressing caspase-1 and GSDMD. A ~100-fold higher amount of the CARD8^{CARD} plasmid was needed to achieve a comparable level of cell death (marked by LDH release) by CARD8^{UPA-CARD}. (g) Titration of NLRP1^{CARD} and NLRP1^{UPA-CARD} with ASC co-expression in HEK293T cells stably expressing caspase-1 and GSDMD. A ~100-fold higher amount of the NLRP1^{CARD} plasmid was needed to achieve a comparable level of cell death (marked by LDH release) by NLRP1^{UPA-CARD}. (h) Model of UPA-enhanced inflammasome formation. The UPA subdomain increases the multivalency of the CARD8 and NLRP1 inflammasomes, lowering the concentration required to nucleate filaments and subsequently signal.

5. Please fix the rotamer outliers in the CARD8 filament.

Responses: We have improved the resolution of the CARD8 map through particle polishing and CTF refinement, and fixed the rotamer outliers (Table S1 and appended PDB validation reports).

6. The outer CARD dimers are likely caused by high concentrations used for filament assembly and reconstruction.

Responses: We now agree with the reviewer on this point. During our revision, we tried titrating the concentration of these dimer mutants in a cellular system and also investigated the ability of the UPA-CARD mutants to form filaments (**Supplementary Fig. 3a-b**, below). While the Y1445A mutant abolished filament formation, the other mutants resembled the wild-type UPA-CARD. Given this new data, the reviewer's suggestion and that the dimerization surface of NLRP1 is only conserved in mammalian NLRP1, but not in more remote NLRP1 or any CARD8, we toned down discussion of the functional significance of NLRP1-CARD dimerization.

Supplementary Figure 3 | The NLRP1 dimer interface has limited impact on inflammasome formation. (a) LDH release when NLRP1 UPA-CARD dimer mutants was titrated in HEK293T cells stably expressing caspase-1 and GSDMD, with ASC co-expression. All mutants behaved similarly as the WT. (b) Filament formation by NLRP1 UPA-CARD dimer mutants in vitro. At the concentration used, most of these mutants formed filaments, like WT (Fig. 3d), except for the Y1445A mutant, which was defective in filament formation, consistent with its impairment in LDH release (Fig. 4d). (c) Multiple sequence alignment (Clustal Omega) between NLRP1^{CARD} and CARD8^{CARD} homologs, coloured with ESPrint⁷³. The NLRP1 dimer interface residues (annotated with purple asterisks) are largely conserved across NLRP1 paralogs in mammals, but not in zebrafish NLRP1 or all CARD8 paralogs.

**We thank Reviewer 3 for their enthusiasm and well as their insights and suggestions.**

**Reviewer #3 (Remarks to the Author):**

The authors present cryoEM structures of two filamentous oligomers of C-terminal portions of
caspase recruitment domains, NLRP1 and Card8, as well as a ASC-caspase1 octamer. The
association of the various domains, as well as their potential orientation relative to ASC and
caspase1 binding partners, are discussed, although the UPA portion of both NLRP1 and Card8
are not evident.

While the work in this paper seems technically sound, the interpretation leaves something
to be desired. The lack of any UPA signal in the structure makes any conclusions drawn about its
function from these structures impossible, and so must rely on other biochemical evidence. This
evidence (activity assays using LDH as a proxy for cell death, and caspase 1 levels) is indirect
evidence of UPA function, and indeed there is very little to demonstrate that UPA is still present -
though I do believe it may well be simply disordered. A western blot against UPA would
demonstrate its presence in a purified fraction. The authors repeatedly discuss UPA dimerisation,
for which I could find no supporting evidence of any sort presented in this manuscript.

Furthermore, the authors present conclusions about the binding interface between CARD
and ASC and Caspase-1 without giving any indication of the experimental procedures. The level
of detail they ascribe to their analyses - namely amino acid level specificity - represents a drastic
overreach of the evidence they present. This is particularly the case when they decide between
model orientations "by visual inspection of the charge complementarity". Again, their analysis is
plausible, but wholly unsupported by structural evidence or documented procedures. I strongly
recommend aiming for a higher level of rigour in their analysis and presentation.

On the whole, I enjoyed the concept of the paper. In particular, I thought the octamer
structure to be revealing in determining orientation of binding. The EM data seem adequate, and,
at first glance, appear to be superior in resolution to the related structures cited in a recent bioRxiv
deposition. The topic is of interest, and is pertinent to the field of inflammasome biology. The
figures are beautifully done, and clear in what they attempt to convey. However, I find the
interpretation of the data lacking in rigour, and overreaching in its scope. I do not recommend
publication of this manuscript in its present form, but I encourage the authors either to introduce
experiments (with experimental details) that support their plausible claims, or formulate an
interpretation of the existing data consistent with the conclusions that can be drawn from the
evidence at hand.

**Responses:** We thank the reviewer for their insights and suggestions and are especially grateful
for their very detailed reading of our paper.

**Concerns and comments:**

**1. Comment:** Fig1 - The FSC plots in e) and h) are not at 1 at 10Å, even though you claim 3.5
and 3.7 Å resolution. This is not good. In general, the falloff of the curve in e) is strange, as the
map-model correlation is better than the half-map correlation. How do you explain this? - You
seem to be using a 0.5 criterion for the map-model. A more conservative criterion seems to be
warranted here given your FSC curves - 0.7 is used in some literature and may be more
appropriate.

**Responses:** We have improved our filament maps with CTF refinement and particle polishing,
and we collected a new dataset for the CARD octamer. All FCS curves correctly converge to 1 at
low resolution and display characteristics as expected (**Fig 1e, 1h** and **Supplementary Fig. 4e,**
below).

Figure 1 | Structure determination of CARD8-CT and NLRP1-CT filaments. ... (d) Cryo-EM 2D classification of CARD8-CT filaments. **(e)** Gold-standard Fourier shell correlation (FSC) and map-model correlation plots of the CARD8-CT filament 3D reconstruction, which gave an overall resolution of 3.3 Å. **(f)** Local resolution of the CARD8-CT filament calculated with RELION's local resolution estimation^{61,72}. **(g)** Cryo-EM 2D classification of NLRP1-CT filaments. **(h)** FSC plots of the NLRP1-CT filament 3D reconstruction, which gave an overall resolution of 3.6 Å. **(i)** Local resolution of the NLRP1-CT filament with RELION's local resolution estimation^{61,72}.

Supplementary Figure 4 | Structure determination of the ASC-caspase-1 octamer. ... (e) Gold-standard FSC and map-model correlation plots of the ASC-caspase-1 octamer, which gave an overall resolution of 3.9 Å.

2. Comment: Page 5 - "One possible explanation is that CARD8UPA does not follow the CARD helical symmetry but is orderly associated with the central CARD8CARD filament." - Card8CARD and UPA are covalently linked. How can UPA be "orderly associated", but not visible in the average? Wouldn't that be disordered? Have you tried focused classification instead of just subtraction and asymmetric classification? Also, your nomenclature is confusing. It sounds like you have two different filaments: "no UPA density was observed in the CARD8-CT filament structure, in which only the CARD8 CARD filament was visible" You have one filament here: CARD8-CT. It's made of a CARD and UPA - one you can see, one you can't.

Responses: The reviewer is right that we have only one filament which is CARD8-CT and that it is composed of CARD and UPA. To elucidate the structure of UPA in the CARD8-CT, we did try focused classification without success. Given all the data we have, we posit that the UPA itself might dimerize or oligomerize along the filament, but its flexible linkage to the CARD causes it to average out during alignment as it assumes stochastic positions relative to the CARD and other UPA dimers/oligomers. We rephrased this language in the text. We also provided additional support of UPA oligomerization (see response to comment 4 below). Further, we fixed inconsistencies throughout in nomenclature, and superscript where appropriate (The CARD8 UPA-CARD fragment = CARD8^{UPA-CARD}, CARD8 CARD alone = CARD8^{CARD}, etc.).

3. Comment: Fig 3 - It seems to me that the octamer in 3a) and the detailed interactions in 3c) are modeling exercises and are unsupported by structural data presented here. This is not very

clear in the text or the figure caption. What is the evidence that ASC or Caspase-1 filaments will
stack on top of Card8? Nothing of the sort is presented.

**Responses:** Specificity of CARD8 for caspase-1 and NLRP1 for ASC was demonstrated
previously (Ball *et al.*, Life Sci Alliance 2020). NLRP1 requires ASC for caspase-1-mediated cell
death, co-localizes with ASC, and pulls down ASC oligomers when DSS crosslinked. CARD8
does not require ASC, does not co-localize with ASC, and does not pull-down ASC when DSS
crosslinked. In a split luciferase assay, it was also evident that CASP1^{CARD} exclusively binds
caspase-1^{CARD}, whereas NLRP1^{CARD} exclusively engages ASC^{CARD}. Upon reviewer's question and
suggestion and due to the known CARD8–caspase-1 interaction and the known NLRP1–ASC
interaction, we now modeled only these potential interfaces to elucidate that the Ib, IIb, IIIb surface
of CARD8 interacts with the Ia, IIa, IIIa surface of caspase-1 (**Fig. 6a**, below), and that the Ib, IIb,
IIIb surface of NLRP1 interacts with the Ia, IIa, IIIa surface of ASC (**Fig. 6c**). By contrast, the
opposite interactions are incompatible (**Fig. 6b, 6d**), and thus in the orientations shown, CARD8
filament stacks on top of the caspase-1 filament and the NLRP1 filament stacks on top of the ASC
filament (**Fig. 6a, 6c**).

We agree that all potential interfaces should be modeled extensively, and while not
presented, we had inspected every permutation of the potential CARD8/ASC/Caspase-1 and
NLRP1/ASC/Caspase-1 complexes (**Figure**, below). What we found is that all the “impossible”
interfaces have predicted steric and/or electrostatic clashes to justify why these interactions do
not occur.

D. P. Ball *et al.*, Caspase-1 interdomain linker cleavage is required for pyroptosis. *Life Sci Alliance*
**3**, (2020).

**Figure 6 | Modelled specific interfaces of the CARD8-CT and NLRP1-CT filaments for**
 **caspase-1 and ASC, respectively.** (a-b) Opposing arrangements of CARD8 and caspase-1 in
 their hypothetical modes of interaction. The electrostatic surfaces indicate that negatively charged
 Type b surface of CARD8 most likely matches the positively charged Type a surface of caspase-
 1 in (a). (c-d) Opposing arrangements of NLRP1 and ASC in their hypothetical modes of
 interaction. The electrostatic surfaces indicate that largely negatively charged Type b surface of
 NLRP1 most likely matches the positively charged Type a surface of ASC in (c). (e) Detailed
 modelled CARD-CARD type I-III interactions between CARD8 (green) and caspase-1 (gold). (f)
 Detailed modelled CARD-CARD type I-III interactions between NLRP1 (green) and ASC (gold).
 (g) Simplified illustration of NLRP1 and CARD8 hierarchy in inflammasome signalling. NLRP1
 recruits ASC, followed by capase-1 recruitment. In contrast, CARD8 can only recruit caspase-1
 directly.

Figure | Structural analysis on the incompatibility of various hypothetical CARD-CARD interactions involving CARD8 and NLRP1, highlighting potential steric and electrostatic clashes (dotted red ovals). (a) Modelled CARD8-Caspase-1 interaction in the reverse direction show in Fig. 6e. (b-c) Modelled CARD8-ASC interactions in two opposite directions. (d) Modelled NLRP1-ASC interaction in the reverse direction show in Fig. 6f. (e-f) Modelled NLRP1-Caspase-1 interactions in two opposite directions.

**4. Comment:** Page 9 - "The functional role of UPA in inflammasome signalling is suggestive of
 its ability to dimerize or oligomerize such that its presence promotes CARD filament formation." -
 The data presented are not suggestive to me of any dimerisation of UPA (which is a feature also
 present in the schematic model in Fig4g). I see no structural or biochemical evidence presented
 here that corroborates the assertion of UPA dimerisation, though CARD dimerisation does seem
 to involve UPA presence.

**Responses:** The reviewer has identified a key lack of experimental evidence in our original
 submission for which we planned to amend in the revision. To investigate whether the UPA
 domain facilitates assembly of the UPA-CARD inflammasome, we titrated NLRP1 and CARD8
 UPA-CARD versus their CARD-alone counterparts for filament formation *in vitro* and LDH release
 in cells, and examined UPA alone for its aggregation property (new **Fig. 3** below). *In vitro*, 0.25
 μM CARD8 or NLRP1 UPA-CARD (**Fig. 3b and d**) induced filament formation; by contrast, only
 at 15 μM CARD8^{CARD} formed filaments (**Fig. 3a**) and even at 30 μM , NLRP1^{CARD} failed to form
 filaments (**Fig. 3c**). In cells, at a lower amount of plasmids transfected (20 ng or 200
 544 ng/transfection), the NLRP1 or CARD8 UPA-CARD induced more LDH release than their CARD
 alone counterparts; only at the highest amount of plasmids transfected (2000 ng/transfection),
 NLRP1 or CARD8 UPA-CARD and CARD induced similar levels of LDH release (**Fig. 3f and g**).
 Finally, purified MBP-fused UPAs from NLRP1 and CARD8 were already oligomers even with the large
 MBP tag that often prevents oligomerization (**Fig. 3e**). Thus, while not absolutely required,
 the UPA subdomains lower the threshold needed for inflammasome signaling due to their ability
 to oligomerize, which is particularly important in physiological contexts when the endogenous
 protein concentrations are low.

filaments, visualized by negative staining EM. Purified monomeric MBP-fused proteins were
cleaved at concentrations ranging from 0.1-15 μM . CARD8^{CARD} filaments did not appear
consistently until 15 μM . CARD8^{UPA-CARD} filaments formed consistently at 0.25 μM . Arrows in (a)
indicates single, unbundled filaments. (c-d) Concentration-dependent formation of NLRP1^{CARD}
(c) and NLRP1^{UPA-CARD} (d) filaments. Purified monomeric MBP-fused proteins were cleaved at
concentrations ranging from 0.1-30 μM . NLRP1^{CARD} filaments did not appear even at 30 μM
concentration, whereas NLRP1^{UPA-CARD} filaments formed consistently at 0.25 μM . (e) MBP-fused
CARD8^{UPA} and NLRP1^{UPA} (25 nM) imaged by negative staining EM, which already showed
oligomers even with the large MBP tag. (f) Titration of CARD8^{CARD} and CARD8^{UPA-CARD} in
HEK293T cells stably expressing caspase-1 and GSDMD. A ~100-fold higher amount of the
CARD8^{CARD} plasmid was needed to achieve a comparable level of cell death (marked by LDH
release) by CARD8^{UPA-CARD}. (g) Titration of NLRP1^{CARD} and NLRP1^{UPA-CARD} with ASC co-
expression in HEK293T cells stably expressing caspase-1 and GSDMD. A ~100-fold higher
amount of the NLRP1^{CARD} plasmid was needed to achieve a comparable level of cell death
(marked by LDH release) by NLRP1^{UPA-CARD}. (h) Model of UPA-enhanced inflammasome
formation. The UPA subdomain increases the multivalency of the CARD8 and NLRP1
inflammasomes, lowering the concentration required to nucleate filaments and subsequently
signal.

Additionally, we recently released a pre-print of our human DPP9-NLRP1 complex structure
(Hollingsworth*, Sharif*, and Griswold* *et. al.*, BioRxiv,
<https://www.biorxiv.org/content/10.1101/2020.08.14.246132v2>), in which we observed an UPA-
UPA oligomerization interface. This interface was also present on the rat DPP9-NLRP1 complex
(Huang* and Zhang* *et. al.*, BioRxiv,
<https://www.biorxiv.org/content/10.1101/2020.08.13.250241v2>). Point mutations of this UPA-
UPA interface abolished NLRP1 inflammasome signaling in cells while maintaining
autoproteolytic functions of the protein, and impaired UPA-CARD filament formation. Together,
these results provide a strong premise for a bona fide functional role of UPA oligomerization in
NLRP1 and CARD8 inflammasomes.

**5. Comment:** Page 9 - "Based on these data and analyses, we propose a model of UPA- induced
NLRP1CARD filament formation in which flexibly linked UPA dimers or oligomers promote the
intrinsic tendency of NLRP1CARD dimerization and its helical polymerization to mediate
inflammasome formation (Fig. 4g)." - And yet you don't see UPA in either structure... can you
hypothesize why? Also, the orientation of CARD dimerisation would necessitate an asymmetry in
the UPA domain on the N-Terminal end of your constructs.

**Responses:** The disordered linker separating these domains exceeds 17 amino acids which
should allow for sufficient flexibility to accommodate most UPA orientations. In our recently
released pre-print (Hollingsworth*, Sharif*, and Griswold* *et. al.*, BioRxiv,
<https://www.biorxiv.org/content/10.1101/2020.08.14.246132v2>), the oligomerization interface
itself involves ordered front-to-back interactions which could be accommodated by adjacent
CARDS or those along the z-axis of the filament given the linker size. If the position of this flexibly
linked UPA dimer/oligomer was stochastic relative to the central CARD helix, it would completely
average out during cryo-EM image processing (**Fig. 3h**, above).

**6. Comment:** How would these differences be bridged in your model? Page 10 - "To further
elucidate the structural basis for the NLRP1-ASC interaction, we analysed the modelled
interfaces type by type." - You do not describe how you arrive at this model. It does not seem to
be based on structural data presented here, and, although plausible, remains speculative. You
go on to give amino-acid level interactions for this model in the absence of structural data, or
methods for how you generated it. This is not acceptable.

**Responses:** We apologize for any oversight in communicating our modelling. We now discuss
how we arrive at these models both in the **Methods** section and in the **Results** section.

**7. Comment:** Page 11 - "and that UPA oligomerization decreases the threshold for filament
assembly by locally concentrating the CARs." What is your evidence of this? I see a
concentration-based test for CARD8, but conspicuously not for NLRP1 where the possibility of
URA dimerisation is much higher. If this is true, shouldn't the filament assembly concentration be
substantially lower?

**Responses:** The reviewer has identified a key lack of experimental evidence in our original
submission for which we planned to amend in the revision. To investigate whether the UPA
domain facilitates assembly of the UPA-CARD inflammasome, we titrated NLRP1 and CARD8
UPA-CARD versus their CARD-alone counterparts for filament formation *in vitro* and LDH release
in cells, and examined UPA alone for its aggregation property (new **Fig. 3** above). *In vitro*, 0.25
μM CARD8 or NLRP1 UPA-CARD (**Fig. 3b and d**) induced filament formation; by contrast, only
at 15 μM CARD8^{CARD} formed filaments (**Fig. 3a**) and even at 30 μM , NLRP1^{CARD} failed to form
filaments (**Fig. 3c**). In cells, at a lower amount of plasmids transfected (20 ng or 200
623 ng/transfection), the NLRP1 or CARD8 UPA-CARD induced more LDH release than their CARD
alone counterparts; only at the highest amount of plasmids transfected (2000 ng/transfection),
NLRP1 or CARD8 UPA-CARD and CARD induced similar levels of LDH release (**Fig. 3f and g**).
Finally, purified MBP-fused UPAs from NLRP1 and CARD8 were already oligomers even with the
large MBP tag that often prevents oligomerization (**Fig. 3e**). Thus, while not absolutely required,
the UPA subdomains lower the threshold needed for inflammasome signaling due to their ability
to oligomerize, which is particularly important in physiological contexts when the endogenous
protein concentrations are low.

**8. Comment:** Page 11 - "In NLRP1-CT, UPA dimerization or oligomerization is made apparent
by an intrinsic propensity for CARD-CARD dimerization along an interface that does not
participate in classical type I, II, or III helical interactions" - again, you are conflating CARD
dimerisation with UPA dimerisation.

**Responses:** We provide evidence of UPA oligomerization (**Fig. 3e**, above) and wrote a clearer
section on how UPA lowers the threshold for filament formation/signaling in the main text (please
also see comment 3).

**Minor points:**

**1. Comment:** Why no line numbers? Boooo to all of you!

**Responses:** We have been thoroughly shamed into adding line numbers on the revised
manuscript despite their objectionable aesthetics, and in this "response to reviewers" document.

**2. Comment:** Abstract: "oligomerized" => oligomerisation?

**Responses:** We apologize for the confusion caused by the omission of a punctuation mark. We
have now added it and the sentence is now: "Here, we report cryo-EM structures of NLRP1-CT
and CARD8-CT assemblies, in which the respective CARs form central helical filaments that
are promoted by oligomerized, but flexibly linked, UPAs surrounding the filaments."

**3. Comments:** Abstract - UPA, ASC, not defined. Page 4 - ASC and UPA still not defined.

**Responses:** We now define all acronyms in the paper. However, we caution the reader against
the full names of these proteins, as they are not very helpful, and they are perhaps even
misleading in our context (e.g. apoptosis-associated speck-like protein containing a CARD here
being involved in pyroptosis, not apoptosis).

**4. Comment:** Abstract "low-resolution 4 ASCCARD–4 caspase-1CARD octamer" - unclear. Do
you mean 4 Å resolution? Or 4 of each molecule? Rephrase.

**Responses:** We have changed the text to reflect the much-improved octamer structure, "Finally,
we engineer and determine the structure of an ASC^{CARD}–caspase-1^{CARD} octamer, which suggests
that ASC uses opposing surfaces for NLRP1, versus caspase-1, recruitment."

**5. Comment:** Page 5 - "By systematically optimizing cleavage conditions, we purified short (~100-
200 nm) filaments that still contained some uncleaved MBP-tagged proteins." - I don't get it. You
optimised cleavage conditions but still have MBP-tags in your filament? What were you optimising?

**Responses:** Under many conditions, either aggregation or incorporation of untagged protein into
the filaments quenches further proteolysis (likely by burying the cleavage site) and limits
elongation. We optimized cleavage conditions (concentration of 3C protease, cleavage
temperature, time) to cut the fusion tag, prevent aggregation, and yield longer filaments. In the
revised manuscript, we included the sentence, "By systematically optimizing cleavage conditions
for filament elongation and to prevent aggregation", within the main text, and clarified this point in
the methods section with, "These cleavage conditions were optimized extensively to produce
elongated filaments, as aggregation and incorporation of uncleaved MBP-tagged protein in the
NLRP1-CT filaments limits further proteolysis of the MBP-tag and filament elongation". Below, we
also demonstrate the quality of the MBP fusion construct at early stages of expression/purification
and after extensive optimization (both after 3C digestion).

**6. Comment:** Fig1 - 1b) is great, and very helpful. Well done.

**Responses:** Thank you for your kind comment!

**7. Comment:** - Your 2D classes in 1d) are surrounded by a poorly-defined haze. Is this MBP?

**Responses:** We believe that this hazy density is either flexibly linked UPA, MBP, or both. We
added a sentence to read, "Either the UPA itself or any residual MBP molecules appear as noisy
density surrounding the filaments in 2D class averages (Fig. 1d)."

**8. Comment:** Fig2d - the detail images are labeled Type I etc, but this label is not in a consistent
location making it hard to find. The colour of this text is also not coordinated with the interaction
type colour in the overview. Alternatively, the border of the detail could be coloured.

**Responses:** We thank the reviewer for pointing this out, as it is important to make these figures
as clear as possible, especially given the distinct meanings of each color. Since all of the views
in (2d) match the orientation of the overview, we moved the Type I/II/III labels to above each figure.
We changed the Ia/Ib/IIa/IIb/IIIa/IIIb to match with the Type I, Type II, and Type III color scheme.

**9. Comment:** Fig 5 - "positive changes" => charges (twice)

**Responses:** Addressed.

 **10. Comment:** Fig 5a - why "ASC or caspase-1" ? Which did you use? If the fit is the same, just
 say which you used and tell us the other fit is identical. If the point is only to show the orientation
 difference between a) and b), then be more explicit about that.

**Responses:** We have combined and re-tooled Fig. 3 and Fig. 5 as a single, new **Fig. 6** for clarity,
 and re-worded the description.

 **11. Comment:** Page 11 - "hieratical" => hierarchical ?
 **Responses:** Addressed throughout, thank you.

 **12. Comment:** Page 11 - "We posit that these CARD filaments are decorated by the flexibly linked
 UPA subdomains ..." Why are you positing this? Is there concern that UPA is not bound? Could
 it be cleaved? Do you have an antibody against the UPA region?

**Responses:** While there is no commercially-available UPA-specific antibody, all samples were
 validated by SDS-PAGE and mass spectrometry following purification and digestion of the MBP
 tag—the UPA-CARD migrates at the appropriate size (**Supplementary Fig. 1a, Fig. 2a**). We also
 excised bands after 3C treatment and prior to structure determination to confirm the
 identity/coverage by mass spectrometry—we now list peptide coverage in **Supplementary Fig.**
 **1a-b and Fig. 2a-b** (below). Finally, UPA-CARD filaments are also much thicker than the CARD-
 alone filaments (**Fig. 3**, see comment 3).

 **Supplementary Figure 1 | Structure determination of the CARD8-CT filament. (a)** SDS-PAGE
 showing purification of the CARD8^{UPA-CARD} oligomer. Some uncleaved protein incorporates into
 the growing filaments and is present in the column void fraction. **(b)** Validation of the CARD8^{UPA-}
 724 ^{CARD} construct by mass spectrometry. Peptides that were detected are underlined.

 **Supplementary Figure 2 | Structure determination of the NLRP1-CT filament. (a)** SDS-PAGE
 showing purification of the NLRP1^{UPA-CARD} oligomer. Some uncleaved protein incorporates into
 the growing filaments and is present in the column void fraction. **(b)** Validation of the NLRP1^{UPA-}
 731 ^{CARD} construct by mass spectrometry. Peptides that were detected are underlined.

REVIEWERS' COMMENTS

Reviewer #1 (Remarks to the Author):

The authors present a revised manuscript that have addressed all my prior concerns. I think this is a manuscript that merits publication in Nature Communications.

Reviewer #2 (Remarks to the Author):

The authors have sufficiently addressed all the comments/concerns.

Reviewer #3 (Remarks to the Author):

In my view, the authors adequately address the concerns raised. The manuscript has improved readability and clarity, and the revised figures seem suited to the changes made.

I recommend publication.